# BOUND BY SEMANTICITY: UNIVERSAL LAWS GOVERNING THE GENERALIZATION-IDENTIFICATION TRADEOFF

**Marco Nurisso**[1], **Jesseba Fernando**[2,3], **Raj Deshpande**[4], **Alan Perotti**[5,12],
**Raja Marjieh**[6], **Steven M. Frankland**[7], **Richard Lewis**[8], **Taylor Whittington Webb**[9],
**Declan Iain Campbell**[10], **Francesco Vaccarino**[1], **Jonathan D. Cohen**[6,10], **Giovanni Petri**[4,11]
[1]Dipartimento di Scienze Matematiche, Politecnico di Torino
[2]Network Science Institute, Northeastern University
[3]Institute for Experiential AI, Northeastern University
[4]NP Lab, Network Science Institute, Northeastern University London
[5]CENTAI Institute
[6]Department of Psychology, Princeton University
[7]Program in Cognitive Science, Dartmouth College
[8]Department of Psychology, University of Michigan
[9]Microsoft Research
[10]Princeton Neuroscience Institute
[11]Department of Physics, Northeastern University
[12]Intesa Sanpaolo AI Research

## ABSTRACT

Intelligent systems must form internal representations that support both broad generalization and precise identification. Here, we show that these two goals are fundamentally in tension with one another. We derive closed-form expressions proving that any model whose representations have a finite semantic resolution, impairing long-range similarity computations, must lie on a universal Pareto front linking its probability of correct generalization $p_S$ and identification $p_I$. We extend this analysis to general input spaces and to parallel processing scenarios, predicting a sharp $1/n$ collapse in the capacity of processing multiple inputs at the same time. A minimal ReLU network reproduces these laws: a resolution boundary emerges during learning, and empirical $(p_S, p_I)$ trajectories closely match the theory for linearly decaying similarity. Finally, we show that the same limits appear in far more complex systems, including a convolutional neural network and state-of-the-art vision–language models, indicating that learned finite-resolution similarity are broad and foundational informational constraints rather than toy-model artifacts. Together, these results provide a precise theory of the generalization–identification tradeoff and clarify how semantic resolution shapes the representational capacity of deep networks and brains alike.

## 1 INTRODUCTION

**Background.** Modern neural networks are surprisingly good at performing a variety of tasks, rivaling and often surpassing human performance. However, they still exhibit striking limitations in their capabilities to process information, often when they need to process multiple objects at the same time (Campbell et al., 2024; Gong and Zhang, 2024; Rahmanzadehgervi et al., 2024; Rane et al., 2024; Zhang and Wang, 2024; Lewis et al., 2022). Similar limitations are also commonly observed in humans when performing working (short-term) memory tasks Miller (1956); Luck and Vogel (1997); Cowan (2001).

Neural networks employ distributed representations (Hinton et al., 1986; Hinton, 1986; Smolensky, 1990) to process inputs. They enable efficient generalization in unseen situations through, for instance, compositionality, but at the same time suffer from the binding problem —the inability to maintain associations between features when processing multiple inputs simultaneously (Roskies, 1999; Greff et al., 2020; Treisman and Gelade, 1980).

Cognitive science offers a rich literature about the ways in which internal representations can help to generalize. The celebrated Shepard's Universal Law of Generalization (Shepard, 1958a; 1987) states that representations should be arranged in the "psychological space" in a structured way, which echoes the real structure of the entities that are represented. This law has received through the years numerous empirical validations and theoretical support (Shepard, 1958b; Sims, 2018; Tenenbaum and Griffiths, 2001; Chater and Vitányi, 2003). This fundamental idea resonates with recent works in neural network interpretability, showing that feature vectors in the latent spaces of large neural networks are often organized in rich geometric structures (Arora et al., 2018; Engels et al., 2024; Liu et al., 2022; Zhong et al., 2023; Shai et al., 2024; Modell et al., 2025).

Frankland et al. (2021) proposed that these two facts —the striking information processing limitations, and the generalization through structured representations— are strongly related, and are at the heart of a fundamental trade-off which puts in tension generalization versus identification of representations.

**Our contribution.** We investigate the fundamental tradeoff between representational fidelity and distinctness under finite semantic resolution. More precisely, we provide:

1. A framework that quantifies the exact Pareto front between identification and similarity performances, demonstrating how finite resolution creates an inescapable tradeoff;

2. Closed-form expressions for this tradeoff across multiple inputs, noise levels, and varying resolutions, revealing a sharp $1/n$ collapse in multi-item ($n$) processing capacity;

3. Empirical validation showing how this resolution boundary self-organizes during neural network training, with empirical trajectories closely following our theoretical predictions;

4. Confirmation that these limits persist across architectures from simple ReLU networks, to CNNs, to vision-language models, establishing emergent finite resolution as a universal constraint rather than a model-specific artifact.

## 2 SETUP

**Stimulus space and similarity functions.** Assume $A$ to be a model processing stimuli coming from a set $S$ the structure of which is encoded by a distance function $d_S$. For example, $S$ can be the space of color hues or days of the week, naturally arranged in a circle, the set of positions of an item in physical space, or more complex topological spaces, such as a torus, or the Klein bottle of natural image patches (Carlsson et al., 2008).

The model processes the stimuli coming from $S$ and builds representations by mapping them into a latent (or psychological) space $M$ with a map $\Phi: S \to M$, which we assume to be a bijection: this induces naturally a distance $d$ on $M$ via $d(x, y) := d_S(\Phi^{-1}(x), \Phi^{-1}(y))$. In $M$, the representations are processed and compared through a non-negative similarity function $g: M \times M \to \mathbb{R}_+$. For example, if $M$ is a vector space, we can choose $g(x, y) = h(\Phi(x)^\top \Phi(y))$ with $h(x) \geq 0 \forall x$. If $h(x) = \exp(-x)$, this encompasses, but is more general than, the standard self-attention mechanism of a transformer (Vaswani et al., 2017) [1].

The specific form of $g$ is not uniquely specified by the distance $d$, allowing for different degrees of "semanticity" (how the metrical structure $d$ is represented by $g$) with significant impacts on model capabilities. Localized functions $g_x := g(x, \cdot)$ reduce interference between representations, permitting more reliable distinction between them and thus accurate simultaneous processing of multiple representations. Conversely, more distributed $g$ can reflect long-range relations of $S$, thus enhancing generalization capabilities, at the cost of potential interference among distinct but nearby stimuli. In the following, corroborated by seminal works in the cognitive psychology literature (Shepard, 1987), we assume for simplicity that $g$ depends only on the distance between the stimuli: $g(x, y) = g(d(x, y))$.

---

[1] Our similarity function includes common ML metrics: cosine similarity in embedding models, dot-product attention in transformers, and implicit similarity in contrastive learning (InfoNCE, triplet loss). While these mechanisms differ in implementation, they all measure semantic relatedness between representations and are subject to the resolution limits we identify in this work.

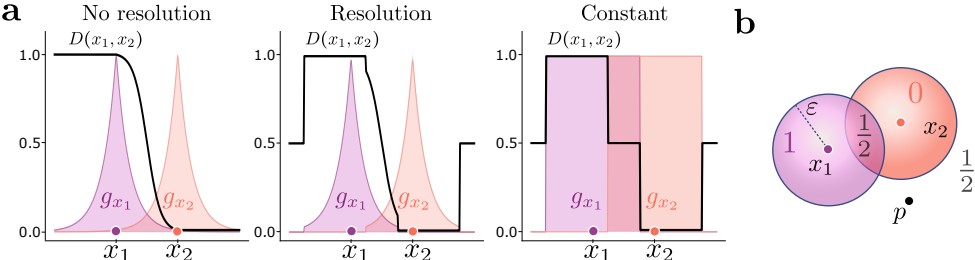

Figure 1: **a.** On the left, exponential similarity functions centered on two stimuli $x_1, x_2 \in M$, with the black line indicating the decision function $g(x_1, p)/(g(x_1, p) + g(x_2, p))$ with no resolution (see Section 2 for explanation). On the center and right, the same quantities are shown in the case of the presence of finite resolution. Notice that the model becomes uncertain for probes far away from stimuli $x_1, x_2$. **b.** Visualization of the constant similarity functions of Definition 1.

**Measures of identification and generalization accuracy.** Following Frankland et al. (2021), we introduce models of two simple tasks that have previously been used to measure identification and generalization accuracy, and that we use in our theoretical analyses below.

We measure the generalization capabilities of $A$ using a *similarity task* in which the model is asked to perform similarity judgments that respect the metric structure of the stimulus space. The model is shown $n$ stimuli $x_1, \ldots, x_n \in S$ and an additional one, called the *probe*, $p \in S$. It is then asked to decide which of the $n$ stimuli is the closest to $p$ according to the distance $d$. Let $(x_1, \ldots, x_n), p$ be sampled independently from $M$ according to a probability measure $\nu$. We call $X$ the random variable encoding the index of the closest item to the probe, i.e. $X = \operatorname*{argmin}_{i=1,\ldots,n} d(x_i, p)$. Intuitively, the decision function represents how the model assesses the evidence when determining which input is most similar to the probe. It formalizes the idea that the model's choice depends on relative similarity strengths rather than absolute values. We call $Y$ the random variable indicating the model's decision, that we model as follows (Luce, 1959):

$$D_i(x_1, \ldots, x_n; p) := \mathbb{P}(Y = i | (x_1, \ldots, x_n, p)) = \frac{g(x_i, p)}{\sum_{k=1}^n g(x_k, p)}. \tag{1}$$

We quantify the overall generalization capability as the probability of the model making the correct decision, i.e. $p_S := \mathbb{P}(Y = X)$.

The *identification* task is used to measure how accurately stimuli can distinguished from one another. The task is the same as the similarity task, but with the exception that the probe is always one of the input stimuli $p \in \{x_1, \ldots, x_n\}$. This will result in the decision function of Equation (1) always being of the form

$$D_i(x_1, \ldots, x_n; x_j) := \mathbb{P}(Y = i | (x_1, \ldots, x_n, x_j)) = \frac{g(x_i, x_j)}{\sum_{k=1}^n g(x_k, x_j)}. \tag{2}$$

If now $X(x_1, \ldots, x_n; x_j) = j$, we write $p_I := \mathbb{P}(Y = X)$ to indicate the probability of the model succeeding in the identification task. Equations (1) and (2) can be interpreted, independent of probabilities, in terms of relative similarity, where $p_S$ is taken to represent the average *relative similarity* of stimuli that are close compared to stimuli that are further apart. In the same way, $p_I$ is the average relative similarity of equal stimuli compared to different stimuli.

Importantly, when $g(x_i, x_j) = \exp(-\mu d(x_i, x_j))$, and the decay rate for the exponential is taken to infinity ($\mu \to \infty$), both $p_S$ and $p_I$ approach 1 (perfect performance); that is, identification and generalization accuracy both benefit by maximizing decay rate. Critically, however, it has been observed empirically that virtually *any* loss of precision (i.e., resolution) in computing the similarity function introduces a fundamental tension –referred to as "Miller's Law" (Frankland et al., 2021)– between $p_S$ (generalization) and $p_I$ (identification accuracy) with respect to decay rate, wherein generalization benefits by *decreases* in decay rate that dramatically degrade identification accuracy (Figure 1a). Here, we provide a formal analysis of this effect, showing that it generalizes to learning in neural networks, where it imposes a fundamental constraint on the interaction between representations and efficiency of processing.

**The effect of resolution.** [2] To show this, we formally consider how a limit in the precision with which the model can compute a similarity function impacts both identification and generalization accuracy. Such a limit might arise from any number of factors: computational noise, finite precision, ReLU activations clamping negative correlations to zero (see Section 4), or imprecisely coded distant relationships. These can all be formalized as a *resolution* $\varepsilon > 0$ such that $g(x, y) \approx \Delta$ if $d(x, y) > \varepsilon$, where $\Delta$ is a noise parameter. As shown in Figure 1a, the resolution drastically affects decision boundaries (the **black** line): for probes sufficiently far from both stimuli, the decision function approaches $1/2$ indicating maximal uncertainty. Resolution thus represents the model's inherent limitation in gauging low similarities between distant stimuli.

**Generalization-Identification Tradeoff (Miller's Law).** To analyze this, we use a simplified similarity function. If $\mathbb{1}_A$ is the indicator function over the set $A$, and $B_r(x)$ is the closed ball of center $x$ and radius $r$ over $M$, $B_r(x) = \{y \in M : d(x, y) \leq r\}$, the similarity function can be defined as follows:

**Definition 1.** *The constant similarity function with resolution $\varepsilon$ and noise $\Delta$ is $g_{\varepsilon;\Delta}(x, y) = \mathbb{1}_{B_\varepsilon(x)}(y) + \Delta \mathbb{1}_{M \setminus B_\varepsilon(x)}(y)$.*

According to this function, the model will judge two things to be similar ($g_{\varepsilon;\Delta}(x, y) = 1$) if and only if they are closer than a certain threshold $\varepsilon > 0$. Outside of this "resolution region" the similarity value is fixed to a noise value $\Delta > 0$.

This simplified model aligns with Shepard's Universal Law of Generalization (Shepard, 1987), where similarity decays exponentially with distance: $g(x, y) = \exp(-\mu d(x, y))$. In Shepard's formulations, the parameter $\mu$ controls the sensitivity to distance, with larger $\mu$ creating sharper similarity boundaries. This is conceptually similar to controlling the temperature parameter in a softmax function, in which lower temperatures induce sharper probability distributions, while higher temperatures make them more uniform. In our framework, $\varepsilon$ serves an analogous role, controlling the distance of the similarity functions or the spatial range of entanglement (or *semanticity*) of the representations. In standard kernel terminology, $\varepsilon$ plays a role akin to a kernel bandwidth, determining how decays with distance. Below, we use this to quantify the generalization-identification tradeoff as a function of $\varepsilon$.

## 3 THEORETICAL RESULTS

We use the constant similarity function defined above to derive closed form solutions for the values of $p_S$ and $p_I$ over a broad class of stimulus spaces and probability distributions over them.

Accordingly, we denote $b_p(\varepsilon)$ as the probability measure of the closed ball of radius $\varepsilon$ centered in $p$, $b_p(\varepsilon) := \nu(B_\varepsilon(p))$. Furthermore, let $\langle b(\varepsilon) \rangle = \mathbb{E}_{p \sim \nu}[b_p(\varepsilon)]$ be the average measure of a ball of radius $\varepsilon$ in $M$, and $\mathrm{Var}(b(\varepsilon))$ its variance. The variance term $\mathrm{Var}(b(\varepsilon))$ captures how the probability mass of $\varepsilon$-balls varies across space. Intuitively, this measures the heterogeneity of the stimulus space, that is, how differently 'crowded' regions are, which, in turn, compromises similarity judgments. Additional assumptions and notations are described in Appendix A.2.

**Theorem 1** (2-item tests). *Let $(M, d, \Sigma, \nu)$ be a separable metric probability space. If, for every $p \in M$, $b_p$ is absolutely continuous on every closed sub-interval of $[0, \infty)$, then, for the noise-free constant similarity function $g = g_{\varepsilon;0}$ it holds that*

$$p_S(\varepsilon) = \frac{1}{2} + \langle b(\varepsilon) \rangle - \langle b(\varepsilon) \rangle^2 - \mathrm{Var}(b(\varepsilon)), \tag{3}$$

$$p_I(\varepsilon) = 1 - \frac{1}{2} \langle b(\varepsilon) \rangle. \tag{4}$$

The proofs can be found in Appendix A.3.

These results have implications for neural architecture design and quantify how much identification performance must be sacrificed to gain generalization ability. These results, being independent of

---

[2] Note on terminology: "resolution" ($\varepsilon$) in this paper strictly refers to the parameter controlling the distance threshold beyond which similarities collapse to noise level $\Delta$. Higher $\varepsilon$ values mean the model preserves similarity information across greater distances.

model choices, provide multiple insights on how $p_S, p_I$ depend on the resolution $\varepsilon$ and on their relation.

First, note that the variance of the ball volume appears in Equation (3) as a term responsible for decreasing the probability of success in the similarity test. This happens when the probability distribution is non-uniform or the space is heterogeneous (as for a manifold with boundary). Spaces which are homogeneous (in Haar measure) with uniform probability distributions will have $\mathrm{Var}(b(\varepsilon)) = 0$, hence performing similarity tests on them will be easier. Therefore, models will perform better on uniform data manifolds (such as rotations), than on manifolds with varying density (such as natural images).

The specific values of $p_I(\varepsilon)$ and $p_S(\varepsilon)$ can vary depending on the space chosen. However, assuming $\mathrm{Var}(b(\varepsilon)) = 0$, they are both parametrized by $\langle b(\varepsilon) \rangle$, which is always a non-decreasing function of $\varepsilon$ from 0 to 1. This means that, in the $(p_S, p_I)$ plane, there is a "universal" Pareto curve relating identification to generalization accuracy that is independent of $M$ and $\nu$ (Figure 2a). Indeed, as we will show in Section 4, the distance of empirical performances from the Pareto front directly quantifies the additional 'difficulty' introduced by the heterogeneity of the stimuli space (Figure 2b).

This curve exhibits three regimes as a function of the ball's resolution $\varepsilon$.

**Low $\varepsilon$ regime.** For small resolutions, the similarity functions act like Dirac deltas, meaning that representations do not interfere with one another and thus are perfectly distinguishable ($p_I \approx 1$). However, small resolutions mean that the model is able to recognize two objects as similar only if they are very close, limiting generalization ($p_S \approx 0.5$, chance level).

**Medium $\varepsilon$ regime.** Increasing $\varepsilon$ elicits the similarity-identification tradeoff: As $\varepsilon$ increases, the similarity measure for more distant stimuli becomes more robust, and thus the structure of the space can be more accurately represented. However, this comes at the cost of nearby stimuli becoming more similar, thereby producing interference that decreases $p_I$. Importantly, $p_S$ reaches a maximum at $\langle b(\varepsilon) \rangle = \frac{1}{2}$, i.e. when the average ball covers half of the space.

**High $\varepsilon$ regime.** Once $\epsilon$ increases beyond $\langle b(\varepsilon) \rangle > \frac{1}{2}$, the cases in which stimuli interfere $d(x_1, p) \leq \varepsilon, d(x_2, p) \leq \varepsilon$ outweigh the ones in which the probe is too far away $d(x_1, p) > \varepsilon, d(x_2, p) > \varepsilon$, resulting in a decrease in both $p_S$ and $p_I$.

**The effect of noise.** The result of Theorem 1 can be readily extended to take into account the presence of nonzero noise outside the resolution region.

**Theorem 2** (Noise). *Under the same assumptions of Theorem 1, for the two-item similarity and identification tests with constant similarity functions $g = g_{\varepsilon;\Delta}$ with noise level $\Delta \geq 0$ it holds that*

$$p_S(\varepsilon, \Delta) = \frac{1}{2} + \frac{1 - \Delta}{1 + \Delta}(\langle b(\varepsilon) \rangle - \langle b(\varepsilon)^2 \rangle), \quad (5)$$

$$p_I(\varepsilon, \Delta) = \frac{2 - (1 - \Delta)\langle b(\varepsilon) \rangle}{2 + 2\Delta}. \quad (6)$$

*Proof.* The proof can be found in Appendix A.4. $\square$

The effect of noise can be appreciated in Figure 2a as a monotonous decrease in both $p_S$ and $p_I$.

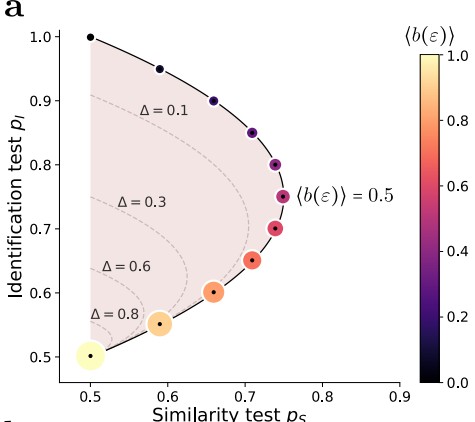

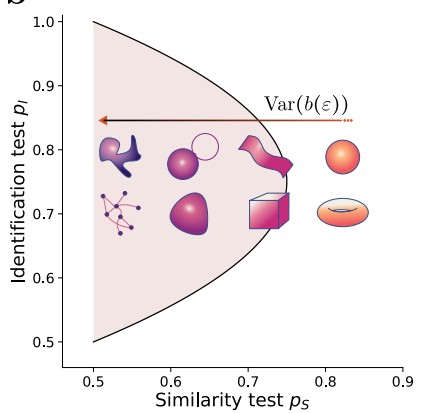

Figure 2: **a.** The region in $(p_S, p_I)$ plane where the model's performances lie (Theorem 1). The black line is parameterized by the resolution $\varepsilon$ and represents the behaviour of the model in homogeneous spaces. **b.** Effect of heterogeneity $\mathrm{Var}(b(\varepsilon))$ on the similarity test performance.

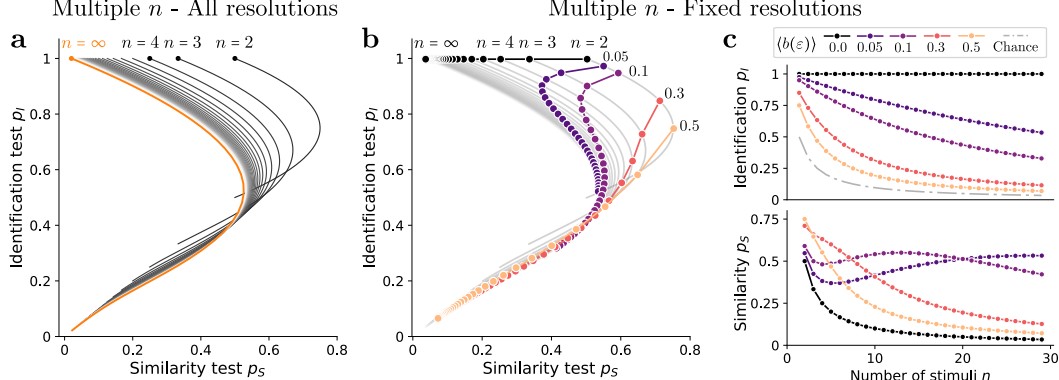

Figure 3: **a.** Similarity-identification curves for different values of $n$ and parameterized by $b_p(\varepsilon) \in [0, 1]$, as described by Equations (7) and (8). **b.** The colored curves correspond to similarity-identification values as the number of inputs $n$ varies, for some fixed values of $b_p(\varepsilon)$. **c.** Similarity (top) and identification (bottom) dependence on $n$ for different resolutions.

**Processing of multiple stimuli.** The foregoing analyses may provide a formal account of why humans and large neural networks alike exhibit dramatic processing constraints in simple tasks (e.g. visual working memory tasks and numerosity judgments), that demand simultaneous processing of multiple stimuli (Campbell et al., 2024). On the one hand, these tasks typically demand generalization (e.g., the processing of stimuli that involve arbitrary combinations of features, such as color, shape and position). On the other hand, performance is typically evaluated based on identification accuracy by identifying individual stimuli. The results above thus suggest that these competing demands run up against the fundamental tension between identification and generalization accuracy, irrespectively of scale or architecture (i.e., even in systems with billions of parameters, such as VLMs or the human brain). When such systems intrinsically value and/or are trained explicitly for generalization, then they will position themselves into the low-medium resolution/semanticity regime ( Figure 2a). Indeed, we can show this is the case by explicitly deriving probabilities of success for $n$-item similarity and identification tasks.

**Theorem 3** ($n$-item tests). *Under the same assumptions of Theorem 1, for the constant noise-free ($\Delta = 0$) similarity function $g = g_{\varepsilon;0}$ we have that*

$$p_S^n(\varepsilon) = \mathbb{E}_{p\sim\nu}\left[\frac{1}{n} + \sum_{k=1}^{n-1} \frac{(1 - b_p(\varepsilon))^{n-k} - (1 - b_p(\varepsilon))^n}{k}\right], \quad (7)$$

$$p_I^n(\varepsilon) = \mathbb{E}_{p\sim\nu}\left[\frac{1 - (1 - b_p(\varepsilon))^n}{n b_p(\varepsilon)}\right]. \quad (8)$$

*Proof.* The proof can be found in Appendix A.5. □

First, note that, despite their apparently complicated formulations, Equations (7) and (8) are polynomials in $b_p(\varepsilon)$ for any fixed $n$ and, given their non-linearity, the expected value over the probes cannot be simplified in general. Thus, for simplicity, we focus on the *homogeneous* case where $b_p(\varepsilon) = b(\varepsilon) \, \forall p \in M$ and $\mathbb{E}$ disappears.

Under this assumption, both similarity and identification performances are once again parameterized by $b(\varepsilon)$, yielding universal pareto curves independent of $M$. Figure 3a shows the shape of the Pareto front for different values of $n$. As a sanity check, note that, as the resolution goes to $b(\varepsilon) = 0$, performance approaches perfect identification for any number of simultaneous inputs with no capacity to generalize $p_S^n(0) = 1/n$ (chance level).

As shown in Figure 3(b,c), the mapping of one curve into the next is not "uniform". For any fixed $\varepsilon > 0$, increasing the number of inputs quickly degrades both identification and generalization performances. Furthermore, Equation (8) shows that for large $n$, $p_I^n(\varepsilon) \approx (b(\varepsilon)n)^{-1}$: identification performance decrease as $1/n$ with a rate given by $b(\varepsilon)$. For a model tasked with learning structured

representations of the input space, and thus optimizing for generalization (say, $b(\varepsilon) \approx 1/2$ for $n = 2$), our analyses predict that the capacity to accurately process multiple representations at the same time will be strongly constrained (Figure 3c).

Interestingly, the bottom panel of Figure 3c shows that the probability of success in the similarity test is non-monotonic in $n$ when $b(\varepsilon)$ is small. Thus, when the model has to deal with a high number of items, it is convenient for it to pick low resolutions. The cost, however, is paid by the significant increase in error for low numbers of items.

These observations provide an elegant explanation for why even large neural network models struggle with multi-object reasoning Campbell et al. (2024): they likely have developed representations that support generalization, but this brings a $1/n$ decrease in identification probability as the number $n$ of objects increase, thus generating the striking capacity limits observed in both humans and large vision-language models. In the next section, we provide empirical evidence that neural networks obey these constraints, first in a simple toy model, and then in multiple large scale networks.

## 4 Toy neural network implementation

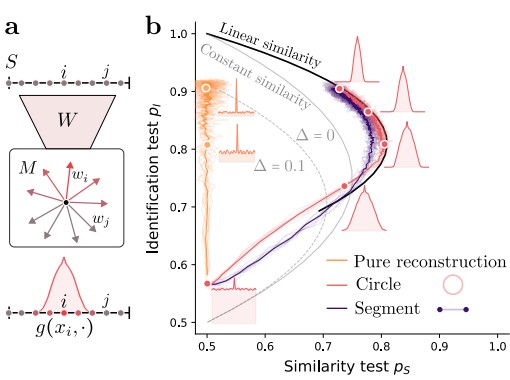

Figure 4: **Emergent resolution and tradeoff in toy architecture.** $(p_S, p_I)$ results for the toy model (**a**) of Section 4 with 50 inputs. **b.** The orange curve shows the average training trajectory for a purely reconstruction loss. The orange insets show the learned (average) similarity function at two epochs. The gray and dashed lines show the curves of Theorem 1 with noise levels $\Delta = 0, 0.1$, respectively. The red curve shows the average training trajectory when the loss is based on the similarity test on a circle while the purple one is trained on stimuli coming from a segment. The black line shows the theoretical performances obtained with linearly decaying similarity functions, as in Proposition 1.

We start from the toy architecture of Elhage et al. (2022), that permits a direct comparison with the analyses above. The input vector $x \in \mathbb{R}_+^l$, whose entries we identify with features, is linearly encoded by $W \in \mathbb{R}^{m \times l}$, decoded by $W^\top$, and then the ReLU activation function $\sigma$ is applied elementwise $f(x) = \sigma(W^\top W x)$ (Figure 4a). When trained with reconstruction MSE loss and sparse inputs, this model displays the phenomenon known as *superposition*: features associated with input dimensions are represented as orthogonally (or *dissimilarly*) as possible to minimize their interference in reconstruction (Elhage et al., 2022). This, in turn, means striving for good identification performance and thus the capability of processing a large number of features simultaneously.

We contrast this with the effect of inducing the model to learn representations with simple forms of metric (semantic) structure. To do so, we consider two spaces of stimuli made of $l$ points $\{x_1, \ldots, x_l\}$ equally spaced in the interval $[0, 1]$: a (flat) circle, with distance $d(x, y) = \min(|x-y|, 1-|x-y|)$, and a segment, with distance $d(x, y) = |x - y|$. The model was trained to perform 3-items similarity tests (as explained in Section 2) on the metric space by encoding its points, the stimuli, as $l$-dimensional one-hot vectors. Given this last assumption, the $i$-th column of $W$, $w_i$, can be interpreted as the latent embedding of $x_i$, and the model's output $f(x_i)_j = \sigma(w_j^\top w_i) := g(x_i, x_j)$ as the non-negative similarity between $x_i$ and $x_j$. The model was trained to convergence 10 times and, for each epoch, we recorded the average similarity and identification ratios $p_S, p_I$ of Equations (1) and (2) using the learned $g$.

Figure 4b shows the resulting training trajectories for three different runs in the similarity-identification plane: the orange run corresponds to trainings with pure reconstruction loss, in red the run with pure similarity task loss on the circle and in purple on the segment. In all cases, we used $l = 50$ stimuli, a hidden dimension of $m = 10$ and repeated the experiment 10 times. See Appendix A.8 for additional details.

As expected, when the network is trained only on reconstruction loss, there is no improvement in $p_S$ but a steady increase in $p_I$. Features are arranged as orthogonally as possible but, due to the low number of hidden dimensions, some interference between them remains. If features are arranged on a line, visualizing the learned similarity function $g(x, \cdot)$ for a fixed $x$ at the last training step shows that it is close to being a Dirac delta on $x$, with smaller-scale random-like noise on other features. Estimating this noise scale $\Delta$ and using that in the equations given by Theorem 2, shows that the corresponding dashed curve accurately predicts the value of $p_I$ at which the training stops.

In contrast, when the network is trained on the semantic task, Figure 4b shows that (starting from the bottom left corner) both $p_S$ and $p_I$ increasing up until the "boundary" is reached, after which similarity begins to decrease. Note that the learned similarity functions $g(x, \cdot)$ for a fixed $x = 0.5$ (the red insets) exhibit a transition from noise to a semantic function that respects the structure of the circle. Furthermore, this structure also exhibits sensitivity to resolution: the model arranges features associated with points further than a certain threshold to have a negative inner product, which is then mapped to zero by the ReLU activation. Moreover, we see that this resolution decreases as training progresses, resulting in an increase of $p_I$ and a decrease in $p_S$.

Not surprisingly, the neural network does not learn constant similarity functions (Section 2), and thus the predictions given by Theorem 1 (in gray) only provide a qualitative prediction. However, the *learned* similarity function $g(x, \cdot)$ appears to be approximately linearly decaying with distance on the circle. Based on this observation, we can analytically derive the values of $p_S$ and $p_I$ for linearly decaying similarities in a circle, finding formulae that approximate Theorem 1.

**Proposition 1** (Linear decay). *On the flat circle $[0, 1]$ with $d(x, y) = \min(|x - y|, 1 - |x - y|)$ sampled with the uniform measure, for the two-item similarity and identification tests with linearly decaying similarity $g(x, y) = \max\left(0, 1 - \frac{d(x,y)}{\varepsilon}\right)$,*

$$p_S(\varepsilon) = \frac{1}{2} + b(\varepsilon) - \left(\frac{3}{2} - \log(2)\right) b(\varepsilon)^2, \quad p_I(\varepsilon) = 1 - (1 - \log(2))b(\varepsilon), \qquad (9)$$

*with $b(\varepsilon) = 2\varepsilon$, $\varepsilon \in [0, 1/2]$.*

The proof can be found in Appendix A.6. Figure 4 shows how the resulting curve (in **black**) provides a good fit to the empirical result. Finally, when the metric space is a segment instead of a circle (purple), the heterogeneity given by the presence of the two endpoints results in an overall reduced $p_S$, as qualitatively predicted by Theorem 1.

## 5 EVIDENCE OF TRADEOFF IN REALISTIC NEURAL NETWORKS

Finally, we summarize experiments and results showing that the effects described above are also observed in networks at scale. We report details on implementations and additional results in Appendix A.8.

**CNNs and evolutionary distance** We fine-tuned a ResNet-50 model (He et al., 2016) to analyze the generalization-identification tradeoff on bird species images (Wah et al., 2011) using a weighted loss function $\mathcal{L} = (1 - \alpha)\,\mathcal{L}_{\text{id}} + \alpha\,\mathcal{L}_{\text{sim}}$, where $\alpha$ controls the bias between identification and generalization. Both tasks employed a triplet design ($x_1$, $x_2$, and $p$): for generalization, the model judged which reference is evolutionarily closer to the probe, using phylogenetic distances as ground truth (Kumar et al., 2022); for identification, it determined the reference species to which the probe belonged. We found that increasing $\alpha$, as a manipulation of similarity, improved generalization while reducing identification accuracy, conforming to the relationships reported above (Figure 5a). Models with higher $\alpha$ values consistently showed enhanced generalization, confirming the ability to manipulate this tradeoff through both training and threshold parameters (see Figure 10 in the SI for the full tradeoff curves as a function of $\varepsilon$ and $alpha$).

**Year similarity task in LLMs.** We then evaluated three open-source large language models (LLMs) (gemma-2b-it (Team et al., 2024), Llama-3.2-3B-Instruct (Grattafiori et al., 2024) and Qwen2.5-7B-Instruct (Yang et al., 2024)) on a similarity task requiring temporal discriminations on the scale of years. The models were prompted to answer questions of the following type "*A was born in $x_1$. B was born in $x_2$. Who was born closest to $p$?*", where A, B are randomized names, a center year $c$ is sampled in $[1500, 1700]$, $x_1 = c - \delta x$, $x_2 = c + \delta x$ for $\delta x \in \{20, 50, 100, 200\}$ and $p = c + \delta p$

takes all years in $[c - 300, c + 300]$. Figure 5b shows the decision curves indicating the empirical probability with which each model responded with the correct answer. This shows that the models' year representations closely follow our assumptions about resolution: all models showed decreased performance as probe dates moved further from reference dates, similar to what we observed with exponentially decaying similarities with noise $g(x_1, x_2) = \exp(-\mu d(x_1, x_2)) + \Delta$ (bottom row).

**Spatial similarity task in VLMs.** Finally, we tested the effects of resolution in two Vision-Language Models (VLMs) (gemma-3-12b-it (Team et al., 2024; Team, 2025a) and Qwen2.5-VL-7B-Instruct (Yang et al., 2024; Team, 2025b)), on a visual spatial similarity task. Four different black shapes were presented to the model in the four corners of the image (Figure 5c), together with a red cross in a random position. The model was tasked with indicating which black shape was closest to the red cross, and we recorded accuracy for each sampled position. Figure 5c shows that, once again, the models display clear resolution limits in their generalization capabilities, similar to those observed in the year task.

# 6   DISCUSSION

We have provided a formal theory of the tradeoff between identification and generalization in systems constrained by finite semantic resolution, building on the formal framework of Frankland et al. (2021). Our closed form expressions reveal a universal Pareto front determined by resolution scale and stimulus geometry, a fundamental limit that is obeyed in empirical tests of model architectures both small and large. Our analysis identifies the optimal resolution for generalization, at which semantic similarity functions tile approximately half of the representational space in discrimination tasks (Sorscher et al., 2022). Beyond this point, increasing resolution impairs identification as representations become too broadly generalized. Below it, representations are discriminable, but fail to capture meaningful similarities, thus compromising generalization. This offers an explanation for why both humans and state-of-the-art neural network models struggle with multi-object reasoning, despite their vast computational resources and remarkable capabilities in other domains.

The spontaneous emergence of this tradeoff across architectures, from minimal ReLU networks to vision-language models, is consistent with our analyses and our empirical findings, that are unified under the hypothesis that finite semantic resolution constitutes an information-theoretic constraint rather than implementation artifact. This, in turn, provides a rigorous mathematical foundation for understanding capacity limits in both artificial and biological systems.

Our theory also indicates how competing representational strategies of intelligent systems are tied to one another: identification demands sharp, distinct representations, while generalization requires coarse, overlapping ones. This tension is echoed in neuroscience literature on *representational efficiency* (coding related items compactly) versus *processing efficiency* (handling multiple items

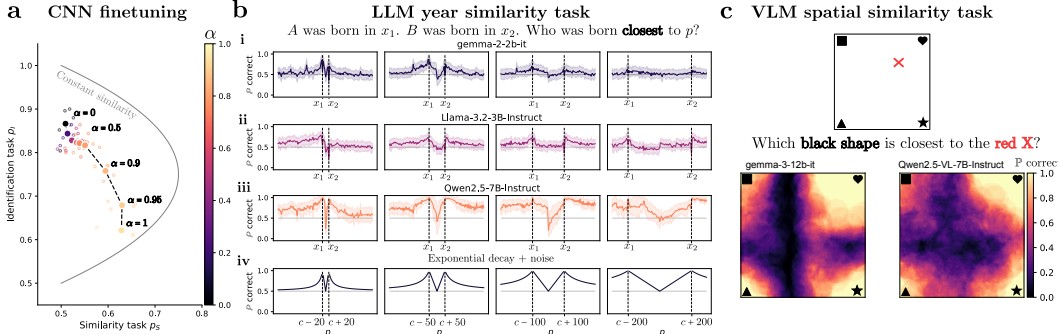

Figure 5: **Empirical resolution tradeoffs across realistic neural architectures**. **a.** A CNN finetuned on bird recognition shows the tradeoff between species identification and generalization to phylogenetic similarity as a function of the weights of generalization $\alpha$ and of the resolution $\epsilon$. **b.** LLMs tasked with comparing years of birth show different regimes of performances, compatible with the existence of an emergent finite resolution ($\sim$ 70–80 years). **c.** VLMs tasked with spatial proximity tasks show decreased accuracy beyond a model-specific resolution scale. Details in appendix A.8.

jointly) (Petri et al., 2024; 2021; Lesnick et al., 2020). Our analyses also provide a formal explanation for empirical observations in neural population coding (Cohen et al., 2020; Ganmor et al., 2015), where semantically clustered "neural thesaurus" structures emerge as optimal strategies under noise constraints, connecting to earlier models of representational redundancy (Curto et al., 2013).

**Limitations and future work.** The present model focuses on non-compositional representations, which do not capture phenomena such as hierarchical syntax, analogical reasoning, or arithmetic—where representations are formed by systematic combinations of simpler parts (Lake and Baroni, 2023; Fodor and Pylyshyn, 1998). Extending our framework to compositional coding schemes remains an important future direction (we provide an initial approach possibility in the SI, see Fig. 6). In addition, while we were able to directly demonstrate the presence of the tradeoff in the toy and CNN models, showing its presence in large language-vision models is still outstanding (despite we provided evidence for finite resolution in them, as also indirectly suggested by Modell et al. (2025).

Future work could further extend our results by: (1) using *synergy–redundancy decompositions* (Proca et al., 2024) to examine how generalization shapes the joint encoding of multiple stimuli; (2) adopting techniques from mechanistic interpretability Bereska and Gavves (2024) to distill the similarity functions directly from internal representations; (3) developing resolution-based diagnostic tools for optimizing neural architectures by targeting task-appropriate generalization-identification balance; and finally (4) testing whether neural manifolds from fMRI or electrophysiology exhibit comparable resolution bounds, potentially establishing semantic resolution as a measurable link between neural geometry and behavioral generalization.

**Reproducibility Statement.** We describe our theoretical framework with complete derivations and provide detailed descriptions of all experimental settings, including architectures, datasets, and training procedures. Hyperparameters, random seed usage, and evaluation protocols are specified in the appendix. Code and data preprocessing scripts to reproduce all results can be found at the repository `https://github.com/nplresearch/generalization`.

### ACKNOWLEDGEMENTS

M.N. acknowledges the project PNRR-NGEU, which has received funding from the MUR – DM 352/2022. G.P. acknowledges partial support by ERC Consolidator Grant RUNES (Grant no. 101171380) and the MSCA Doctoral Network BeyondTheEdge (Grant no. 101120085). F.V. acknowledges that this study was carried out within the FAIR - Future Artificial Intelligence Research and received funding from the European Union Next-GenerationEU (PIANO NAZIONALE DI RIPRESA E RESILIENZA (PNRR) – MISSIONE 4 COMPONENTE 2, INVESTIMENTO 1.3 – D.D. 1555 11/10/2022, PE00000013). J.D.C. was supported by a Vannevar Bush Faculty Fellowship administered by the Office of Naval Research.

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

# A APPENDIX / SUPPLEMENTAL MATERIAL

## A.1 LLM USAGE

LLMs (ChatGPT) were used to aid in polishing the paper after writing.

## A.2 TECHNICAL DETAILS

In this section, we formalize the technical aspects and assumptions required for the results of the paper.

We assume $(M, d, \Sigma, \nu)$ to be a separable metric measure space equipped with the standard metric space topology, the Borel $\sigma$-algebra $\Sigma$ generated by balls in $M$ and with a probability measure $\nu$. This measure $\nu$, which is such that $\nu(M) = 1$, determines how we are sampling stimuli from the stimulus space $M$.

In the following derivations, we will make use of two objects:

- $b_p(\varepsilon) = \nu(B_\varepsilon(p))$, the measure of the ball of radius $\varepsilon$ centered in $p$;

- $S_p$, the push-forward measure on $[0, \infty]$ of the distance function in $p$, $d_p(\cdot) = d(p, \cdot)$ i.e., for any measurable subset of $\mathbb{R}$, $S_p(E) = \nu(d_p^{-1}(E))$.

Note that $b_p$ is the cumulative distribution function of $S_p$ as $S_p((-\infty, \varepsilon]) = S_p([0, \varepsilon]) = b_p(\varepsilon)$ and therefore it is non-decreasing. We also have that $b_p(\infty) := \lim_{\epsilon \to \infty} b_p(\varepsilon) = 1$.

We assume that $b_p$ is an absolutely continuous function on every closed sub-interval of $[0, \infty)$ i.e. such that for every $\epsilon > 0$ there exists $\delta > 0$ such that for any finite set of disjoint intervals $(\alpha_1, \beta_1), \ldots, (\alpha_N, \beta_N)$

$$\sum_{i=1}^{N} (\beta_i - \alpha_i) < \delta \implies \sum_{i=1}^{N} (b_p(\beta) - b_p(\alpha)) < \epsilon.$$

By Nielsen (1997) (Theorem 20.10), the absolute continuity of $b_p(\varepsilon)$ implies that $S_p$ is an absolutely continuous measure w.r.t. the Lebesgue measure $\mu$. This implies, by Radon-Nikodym theorem, that $S_p$ admits a density $f$, $S_p = \int f d\mu$, with $f(\varepsilon) = b_p'(\varepsilon)$ almost everywhere. In fact, we can think of absolute continuity as a stronger notion of continuity, as the fundamental theorem of calculus for Lebesgue integrals (Folland (1999), Theorem 3.35) tells us that, on every interval $[c, d]$, $b_p$ is almost everywhere differentiable, and $b_p(\varepsilon) - b_p(c) = \int_c^\varepsilon b_p'(r) d\mu(r)$.

We now see how these assumptions allow us to notably simplify the derivations of the probability of success in both similarity and identification tasks, while still not being too restrictive. In fact, most non-pathological cases of interest, like probability distributions with differentiable densities on manifolds, satisfy the assumption.

**Lemma 1.** *Let $X$ be the random variable of the correct answer to the $n$-item similarity or identification task $X = \operatorname{argmin} \{d(x_1, p), \ldots, d(x_n, p)\}$. If, for every $p \in M$, $b_p$ is absolutely continuous on every closed interval $[c, d] \subseteq [0, \infty)$, then $\mathbb{P}(|X| > 1) = 0$.*

*Proof.* Let $|X|$ be the cardinality of the set $X$.

$$\mathbb{P}(|X| > 1) = \sum_{k=2}^{n} \binom{n}{k} \mathbb{P}(d(x_1, p) = \cdots = d(x_k, p), d(x_{k+1}, p) > d(x_1, p), \ldots, d(x_n, p) > d(x_1, p))$$
(1)

$$\leq \sum_{k=2}^{n} \binom{n}{k} \mathbb{P}(d(x_1, p) = \cdots = d(x_k, p)) \leq \sum_{k=2}^{n} \binom{n}{k} \mathbb{P}(d(x_1, p) = d(x_2, p))$$
(2)

$$= \sum_{k=2}^{n} \binom{n}{k} \int_0^{\infty} \mathbb{P}(d(x_1, p) = d(x_2, p) = r) d\mu(r) \leq \sum_{k=2}^{n} \binom{n}{k} \int_0^{\infty} \mathbb{P}(d(x_1, p) = r) d\mu(r)$$
(3)

$$= \sum_{k=2}^{n} \binom{n}{k} \int_0^{\infty} S_p(\{r\}) d\mu(r) = 0,$$
(4)

where the last equality comes from the absolute continuity of $S_p$ w.r.t. $\mu$, as $\mu(\{r\}) = 0$. $\qquad\square$

This result tells us that, under the assumptions, there is a probability of $0$ that there are multiple correct answers to the similarity and identification tasks. Therefore, in the following derivations we will always only have to deal with the case $|X| = 1$.

### A.3 PROOF OF THEOREM 1

#### A.3.1 SIMILARITY TASK

*Proof.* Let us derive the probability of succeeding in the similarity task in the case of 2 items. The following proof will be a subcase of the more general one for $n$ items but, given its complexity, it is useful to analyze this subcase separately.

Let $x_1, x_2, p$ be sampled independently from $M$ according to the probability measure $\nu$. Let $X(x_1, x_2, p) = \underset{i \in \{1,2\}}{\arg\min} \, d(x_i, p)$. Notice that $X$ can have three different values

$$\begin{cases} X(x_1, x_2, p) = \{1\} & \text{if } d(x_1, p) < d(x_2, p) \\ X(x_1, x_2, p) = \{2\} & \text{if } d(x_2, p) < d(x_1, p) \\ X(x_1, x_2, p) = \{1, 2\} & \text{if } d(x_1, p) = d(x_2, p). \end{cases}$$

In this last case, when $x_1, x_2$ are equidistant from $p$, any answer to the task will be correct. By Lemma 1, we only need to focus on the first two as the probability that more than one answer is correct is $0$.

We have that

$$\mathbb{P}(Y = X) = \sum_{i=1}^{2} \mathbb{P}(Y = i | X = i) \mathbb{P}(X = i).$$
(5)

Let us now rewrite the probability $\mathbb{P}(Y = i | X = i)$ by conditioning over all possible results of the samplings of $x_1, x_2$ and the probe $p$.

For this, given the independence assumption, we assume that the event $(x_1, x_2, p)$ is an element of the measure space $(M^3, \nu^{\otimes 3})$ equipped with the standard product measure.

$$\mathbb{P}(Y = i | X = i) = \int_{M^3} \mathbb{P}(Y = i | X = i, (x_1, x_2, p)) dP(x_1, x_2, p | X = i),$$

where $dP(x_1, x_2, p | X = i)$ is the conditional measure of the sampling of $x_1, x_2, p$ given the event that $X = i$ which, by Bayes theorem, can be rewritten as

$$dP(x_1, x_2, p | X = i) = \frac{\mathbb{1}[X(x_1, x_2, p) = i] d\nu(x_1) d\nu(x_2) d\nu(p)}{\mathbb{P}(X = i)},$$

where $\mathbb{1}[X(x_1, x_2, p) = i]$ coincides with the conditional law of the (deterministic) variable $X | (x_1, x_2, p)$.

Replacing this in Equation (5) we get

$$\mathbb{P}(X = Y) = \sum_{i=1}^{2} \int_{M^3} \mathbb{P}(Y = i | X = i, (x_1, x_2, p)) \mathbb{1}[X(x_1, x_2, p) = i] d\nu(x_1) d\nu(x_2) d\nu(p) \quad (6)$$

The independence of the samplings of $x_1$ and $x_2$ means that all indices are equally likely to be the correct answer $\mathbb{P}(Y = i | X = i) = \mathbb{P}(Y = j | X = j) \ \forall i, j \in \{1, 2\}$.

$$\mathbb{P}(X = Y) = 2 \int_{M^3} \mathbb{P}(Y = 1, X = 1, (x_1, x_2, p)) \mathbb{1}[X(x_1, x_2, p) = 1] d\nu(x_1) d\nu(x_2) d\nu(p) \quad (7)$$

$$= 2 \int_M \int_M \int_{x_2 \in M: d(x_2, p) > d(x_1, p)} \frac{g(x_1, p)}{g(x_1, p) + g(x_2, p)} d\nu(x_2) d\nu(x_1) d\nu(p) \quad (8)$$

Given the fact that we are considering constant similarity functions $g(x, y) = g_{\varepsilon;0}(x, y)$ which depend only on the distance between $x$ and $y$, $g(x, y) = g(d(x, y))$, we perform the following change of coordinates $d(x_1, p) \mapsto r_1, d(x_2, p) \mapsto r_2$,

$$\mathbb{P}(X = Y) = 2 \int_M d\nu(p) \int_{[0, \infty]} dS_p(r_1) \int_{(r_1, \infty]} dS_p(r_2) \frac{g(r_1)}{g(r_1) + g(r_2)}, \quad (9)$$

where $S_p$ is the pushforward measure induced by the distance function from the probe $p$.

We decompose Equation (9) into two cases: **a.** when $r_1 > \varepsilon$ and thus both items fall outside the resolution region of the probe $p$, and **b.** when $r_1 \leq \varepsilon$ and thus the closest item falls inside.

**a.** In the first case, given that $r_2 > r_1$, we will have that both $x_1$ and $x_2$ are too far from the probe to be recognized as similar, resulting in both numerator and denominator in Equation (9) to be 0. Here we adopt the convention $0/(0 + 0) = 1/2$ to describe the model being maximally uncertain in its decision.

$$2 \int_M d\nu(p) \int_{(\varepsilon, \infty]} dS_p(r_1) \int_{(r_1, \infty]} \frac{1}{2} dS_p(r_2) = \int_M d\nu(p) \int_{(\varepsilon, \infty]} \int_{(r_1, \infty]} dS_p(r_1) dS_p(r_2)$$

To compute this integral, we leverage the almost-everywhere differentiability of $b_p$ and apply the fundamental theorem of calculus

$$\int_{(\varepsilon, \infty]} \int_{(r_1, \infty]} dS_p(r_1) dS_p(r_2) = \int_{(\varepsilon, +\infty]} (1 - b_p(r_1)) dS_p(r_1) \quad (10)$$

$$= \int_{(\varepsilon, +\infty]} (1 - b_p(r_1)) b_p'(r_1) d\mu(r_1) = \left[ -\frac{(1 - b_p(r_1))^2}{2} \right]_\varepsilon^\infty = \frac{(1 - b_p(\varepsilon))^2}{2}. \quad (11)$$

**b.** When $r_1 \leq \varepsilon$ the first item will be considered to be similar to the probe $g(r_1) = 1$, while the second can be both similar and dissimilar.

$$2 \int_M d\nu(p) \int_{[0, \varepsilon]} dS_p(r_1) \int_{(r_1, \infty]} dS_p(r_2) \frac{1}{1 + g(r_2)} \quad (12)$$

$$= \underbrace{2 \int_M d\nu(p) \int_{[0, \varepsilon]} dS_p(r_1) \int_{(r_1, \varepsilon]} \frac{1}{2} dS_p(r_2)}_{\textbf{I}} + \underbrace{2 \int_M d\nu(p) \int_{[0, \varepsilon]} dS_p(r_1) \int_{(\varepsilon, \infty]} dS_p(r_2)}_{\textbf{II}}. \quad (13)$$

The term **I**, just like above, can be computed in the following way

$$\int_M d\nu(p) \int_{[0, \varepsilon]} dS_p(r_1) \int_{(r_1, \varepsilon]} dS_p(r_2) \quad (14)$$

$$= \int d\nu(p) \int_{[0, \varepsilon]} (b_p(\varepsilon) - b_p(r_1)) b_p'(r_1) d\mu(r_1) \quad (15)$$

$$= \int_M \frac{b_p(\varepsilon)^2}{2} d\nu(p) \quad (16)$$

The term **II** is simply given by $2 \int_M b_p(\varepsilon)(1 - b_p(\varepsilon)) d\nu(p)$.

Summing together **a.** and **b.** we arrive at the following:

$$\mathbb{P}(Y = X) = \int_M \frac{1}{2}(1 - b_p(\varepsilon))^2 + \frac{1}{2} b_p(\varepsilon)^2 + 2 b_p(\varepsilon)(1 - b_p(\varepsilon)) d\nu(p) \tag{17}$$

$$= \int_M \frac{1}{2} + b_p(\varepsilon) - b_p(\varepsilon)^2 d\nu(p). \tag{18}$$

We obtain the formula for the probability of succeeding the similarity task:

$$\mathbb{P}(Y = X) = \frac{1}{2} + \langle b(\varepsilon) \rangle - \langle b(\varepsilon)^2 \rangle. \tag{19}$$

$\square$

### A.3.2 IDENTIFICATION TASK

*Proof.* The identification task can be seen as a subset of the similarity task, in which the probe is uniformly picked among the input stimuli. This means that the correct response will be $X(x_1, x_2, p) = \{i \in \{1, 2\} : x_i = p\}$ and both answers will be correct only in the case that $x_1 = x_2$.

Retracing the first steps outlined in Appendix A.3.1, we find that the probability of the model being correct will be

$$\mathbb{P}(Y = X) = 2 \int_M \sum_{p \in \{x_1, x_2\}} \frac{1}{2} \mathbb{P}(Y = 1 | X = 1, (x_1, x_2, p)) \mathbb{1}[X(x_1, x_2, p) = 1] d\nu(x_1) d\nu(x_2) \tag{20}$$

$$= \int_{M^2} \frac{g(x_1, x_1)}{g(x_1, x_1) + g(x_2, x_1)} d\nu(x_1) d\nu(x_1). \tag{21}$$

Note that we used the fact that $p = x_1$ with probability $1/2$ and $p = x_2$ with probability $1/2$. Given that $g(x, x) = 1$ and, by the definition of metric space, $d(x, y) = 0 \iff x = y$, we change coordinates $d(x_1, x_2) \mapsto r$ and rewrite Equation (20) as

$$\mathbb{P}(X = Y) = \int_M d\nu(x_1) \int_{(0,\infty]} \frac{1}{1 + g(r)} dS_{x_1}(r). \tag{22}$$

When $r > \varepsilon$, the second item does not interfere with the probe $p = x_1$ and the model will choose $x_1$ with certainty, while, if $r \leq \varepsilon$, $g(r) = 1$ and it will instead be maximally uncertain.

$$\int_M d\nu(x_1) \int_{(0,\infty]} \frac{1}{1 + g(r)} dS_{x_1}(r) = \int_M d\nu(x_1) \int_{(0,\varepsilon]} \frac{1}{2} dS_{x_1}(r) + \int d\nu(x_1) \int_{(\varepsilon,\infty]} 1 dS_{x_1}(r). \tag{23}$$

$$= \frac{1}{2} \int_M b_{x_1}(\varepsilon) d\nu(x_1) + \int_M (1 - b_{x_1}(\varepsilon)) d\nu(x_1) = 1 - \frac{1}{2} \langle b(\varepsilon) \rangle. \tag{24}$$

$\square$

### A.4 PROOF OF THEOREM 2

The proof proceeds by retracting the proof of the noiseless case with some adjusted constants.

We start from the similarity task success probability, as rewritten in Equation (9). Once again, the integral can be decomposed into two cases: **a.** when $r_1 > \varepsilon$ and thus both items fall outside the resolution region of the probe $p$ and **b.** when $r_1 \leq \varepsilon$ and thus the closest item falls inside.

**a.** When $r_2 > r_1 > \varepsilon$, we have that $g(r_1) = g(r_2) = \Delta$ and thus the ratio $g(r_1)/(g(r_1) + g(r_2)) = 1/2$ resulting in the same term of Equation (11) $(1 - b_p(\varepsilon))^2/2$.

**b.** When $r_1 \leq \varepsilon$ and $r_2 \leq \varepsilon$ both items are similar to the probe and thus we get the same contribution of the term $b_p(\varepsilon)^2/2$ in Equation (16).

The only difference from the proof of the noiseless case is when $r_1 \leq \varepsilon$ and $r_2 > \varepsilon$. In this case, the first item is similar to the probe while the second is not, but the noise erodes the probability of the model picking the first item. Therefore we get the following contribution to $\mathbb{P}(Y = X)$.

$$2 \int_M d\nu(p) \int_{[0,\varepsilon]} dS_p(r_1) \int_{(\varepsilon,\infty]} dS_p(r_2) \frac{1}{1+\Delta} = \frac{2}{1+\Delta} \int_M b_p(\varepsilon)(1 - b_p(\varepsilon)) d\nu(p).$$

Putting all the terms together we get

$$\mathbb{P}(Y = X) = \int_M \frac{1}{2}(1 - b_p(\varepsilon))^2 + \frac{1}{2} b_p(\varepsilon)^2 + \frac{2}{1+\Delta} b_p(\varepsilon)(1 - b_p(\varepsilon)) d\nu(p) \quad (25)$$

$$= \int_M \frac{1}{2} + \left( \frac{2}{1+\Delta} - 1 \right)(b_p(\varepsilon) - b_p(\varepsilon)^2) d\nu(p) \quad (26)$$

$$= \frac{1}{2} + \frac{1-\Delta}{1+\Delta}(\langle b_p(\varepsilon) \rangle - \langle b_p(\varepsilon)^2 \rangle) \quad (27)$$

For the identification task, we start from Equation (22). Now, when $r > \varepsilon$, the second item is outside of the resolution region of $x_1$ but the noise will still make the model's decision not certain.

$$\mathbb{P}(Y = X) = \int_M d\nu(x_1) \int_{(0,\infty]} \frac{1}{1+g(r)} dS_{x_1}(r) \quad (28)$$

$$= \int_M d\nu(x_1) \int_{(0,\varepsilon]} \frac{1}{2} dS_{x_1}(r) + \int d\nu(x_1) \int_{(\varepsilon,\infty]} \frac{1}{1+\Delta} dS_{x_1}(r). \quad (29)$$

$$= \frac{1}{2} \int_M b_{x_1}(\varepsilon) d\nu(x_1) + \frac{1}{1+\Delta} \int_M (1 - b_{x_1}(\varepsilon)) d\nu(x_1) = \frac{1}{1+\Delta} - \frac{1-\Delta}{2(1+\Delta)} \langle b(\varepsilon) \rangle. \quad (30)$$

## A.5 PROOF OF THEOREM 3

**Lemma 2.**

$$\sum_{j=1}^n \binom{n}{j} \frac{1}{j} x^j (1-x)^{n-j} = \sum_{j=1}^n \frac{(1-x)^{n-j} - (1-x)^n}{j}.$$

*Proof.* Let us call $f_n$ the left-hand side of the identity and $g_n$ the right-hand side. We prove the result by showing that the generating functions of the series $(f_n)_n, (g_n)_n$ are equal.

Let us start with $(f_n)_n$.

$$F(z) = \sum_{n=0}^\infty f_n z^n = \sum_{n=0}^\infty \sum_{j=1}^n \binom{n}{j} \frac{1}{j} x^j (1-x)^{n-j} z^n = \sum_{j=1}^\infty \sum_{n=j}^\infty \binom{n}{j} \frac{1}{j} x^j (1-x)^{n-j} z^n \quad (31)$$

$$= \sum_{j=1}^\infty \frac{x^j}{j} \left( \sum_{n=j}^\infty \binom{n}{j} (1-x)^{n-j} z^n \right) = \sum_{j=1}^\infty \frac{x^j}{j} \left( \sum_{k=0}^\infty \binom{j+k}{j} (1-x)^k z^{k+j} \right) \quad (32)$$

$$= \sum_{j=1}^\infty \frac{x^j}{j} z^j (1 - z + xz)^{-j-1} = \frac{1}{(1-z+xz)} \sum_{j=1}^\infty \frac{1}{j} \left( \frac{xz}{1-z+xz} \right)^j \quad (33)$$

$$= -\frac{\log\left(1 - \frac{xz}{1-z+xz}\right)}{1-z+xz} = -\frac{\log\left(\frac{1-z}{1-z+xz}\right)}{1-z+xz}, \quad (34)$$

where we used the generating function identity for the binomial, see (Graham et al., 1989) (Equation 5.56) and the power series expansion of $\log(1-x)$.

Let us proceed in the same way for $g_n$:

$$G(z) = \sum_{n=0}^{\infty} g_n z^n = \sum_{n=0}^{\infty} \sum_{j=1}^{n} \frac{1}{j}((1-x)^{n-j} - (1-x)^n) z^n \tag{35}$$

$$= \sum_{j=1}^{\infty} \frac{1}{j} \left( \sum_{n=j}^{\infty} (1-x)^{n-j} z^n - \sum_{n=j}^{\infty} (1-x)^n z^n \right) \tag{36}$$

$$= \sum_{j=1}^{\infty} \frac{1}{j} \left( (1-x)^{-j} \sum_{n=j}^{\infty} (z-zx)^n - \sum_{n=j}^{\infty} (z-zx)^n \right) \tag{37}$$

$$= \sum_{j=1}^{\infty} \frac{1}{j} \left( (1-x)^{-j} \frac{(z-zx)^j}{1-z+zx} - \frac{(z-zx)^j}{1-z+zx} \right) \tag{38}$$

$$= \frac{1}{1-z+zx} \left( \sum_{j=1}^{\infty} \frac{1}{j} \left( \frac{z-zx}{1-x} \right)^j - \sum_{j=1}^{\infty} \frac{1}{j} (z-zx)^j \right) \tag{39}$$

$$= \frac{-\log\left(1 - \frac{z-zx}{1-x}\right) + \log(1-z+zx)}{1-z+xz} = -\frac{\log\left(\frac{1-z}{1-z+xz}\right)}{1-z+xz} \tag{40}$$

$\square$

### A.5.1 SIMILARITY TASK

Recall that in the $n$-item similarity task, we are sampling independently $n$ stimuli $x_1, \ldots, x_n$ and a probe $p$ and we ask the model to find which among the stimuli is the closest to $p$. Recall that Lemma 1 tells us that the probability of having more than a correct answer is 0.

Retracing the first steps in Appendix A.3, we find that

$$\mathbb{P}(Y = X) = \sum_{i=1}^{n} \mathbb{P}(Y = i | X = i) \mathbb{P}(X = i) \tag{41}$$

$$\tag{42}$$

By the symmetry induced by the independence of the sampling, we see that $\mathbb{P}(Y = i | X = i) = \mathbb{P}(Y = j | X = j) = \mathbb{P}(Y = 1 | X = 1) \; \forall i, j$ and $P(X = i) = P(X = 1) \; \forall i$, and thus we can restrict to the case when the closest stimulus is the first one.

$$\mathbb{P}(Y = 1 | X = 1) = \int_{M^n \times M} \mathbb{P}(Y = 1 | X = 1, (x_1, \ldots, x_n, p)) dP(x_1, \ldots, x_n, p | X = 1) \tag{43}$$

$$= \int_{M^n \times M} \mathbb{P}(Y = 1 | X = 1, (x_1, \ldots, x_n, p)) \frac{\mathbb{1}[X(x_1, \ldots, x_n, p) = 1]}{\mathbb{P}(X = 1)} d\nu(x_1) \cdots d\nu(x_n) d\nu(p). \tag{44}$$

$$\mathbb{P}(Y = X) \tag{45}$$

$$= n \int_{M^n \times M} \mathbb{P}(Y = 1 | X = 1, (x_1, \ldots, x_n, p)) \mathbb{1}[X(x_1, \ldots, x_n, p) = 1] d\nu(x_1) \cdots d\nu(x_n) d\nu(p) \tag{46}$$

$$= n \int_M d\nu(p) \int_M d\nu(x_1) \int_{d(x_2,p)>d(x_1,p)} \cdots \int_{d(x_n,p)>d(x_1,p)} \frac{g(x_1, p)}{\sum_{i=1}^n g(x_i, p)} d\nu(x_2) \cdots d\nu(x_n). \tag{47}$$

Given that $g$ is a constant similarity function $g(x, y) = g_{\varepsilon;0}(x, y)$ which depends only on the distance between $x$ and $y$, we perform the change of coordinates $d(x_i, p) \mapsto r_i$ with $S_p$ being the pushforward measure induced by the distance function from the probe $p$.

$$\mathbb{P}(X = Y) = n \int_M d\nu(p) \int_{[0,\infty]} dS_p(r_1) \int_{(r_1,\infty]} dS_p(r_2) \cdots \int_{(r_1,\infty]} dS_p(r_n) \frac{g(r_1)}{\sum_{i=1}^n g(r_i)}. \tag{48}$$

We now consider two cases separately. **a.** If $r_1 > \varepsilon$, no item falls close enough to the probe and thus $g(x_i, p) = 0 \; \forall i = 1, \ldots, n$ and the model's response is random:

$$n \int_M d\nu(p) \int_{(\varepsilon,\infty]} dS_p(r_1) \int_{(r_1,\infty]} dS_p(r_2) \cdots \int_{(r_1,\infty]} dS_p(r_n) \frac{1}{n} \tag{49}$$

$$= \int_M d\nu(p) \int_{(\varepsilon,\infty]} (1 - b_p(r_1))^{n-1} dS_p(r_1) = \int_M d\nu(p) \int_{(\varepsilon,\infty]} (1 - b_p(r_1))^{n-1} b'_p(r_1) d\mu(r_1) \tag{50}$$

Notice now that $b_p$ absolutely continuous implies that $(1 - b_p(r_1))^{n-1}$ is absolutely continuous and, by Lebesgue's theorem, it is differentiable almost everywhere and the fundamental theorem of calculus holds (see Appendix A.2). We thus deduce that

$$\int_M d\nu(p) \int_{(\varepsilon,\infty]} (1 - b_p(r_1))^{n-1} b'_p(r_1) d\mu = \int_M d\nu(p) \left[ -\frac{(1 - b_p(r_1))^n}{n} \right]_\varepsilon^\infty \tag{51}$$

$$= \int_M \frac{(1 - b_p(\varepsilon))^n}{n} d\nu(p). \tag{52}$$

**b.** If $r_1 \leq \varepsilon$, then the closest stimulus is similar to the probe $g(r_1) = 1$ and we write

$$n \int_M d\nu(p) \int_{[0,\varepsilon]} dS_p(r_1) \int_{(r_1,\infty]} dS_p(r_2) \cdots \int_{(r_1,\infty]} dS_p(r_n) \frac{1}{1 + \sum_{i=2}^n g(r_i)}. \tag{53}$$

Each item $i > 1$ can fall either inside of $B_\varepsilon(p)$ and contribute to the denominator of the decision function, or fall outside. Given that the denominator only depends on the *number* of stimuli which fall in $B_\varepsilon(p)$ and not on their index, we can write Equation (53) as

$$n \int_M d\nu(p) \int_{[0,\varepsilon]} dS_p(r_1) \sum_{k=0}^{n-1} \binom{n-1}{k} (b_p(\varepsilon) - b_p(r_1))^k (1 - b_p(\varepsilon))^{n-1-k} \frac{1}{k+1} \tag{54}$$

$$= n \int_M d\nu(p) \sum_{k=0}^{n-1} \binom{n-1}{k} (1 - b_p(\varepsilon))^{n-1-k} \frac{1}{k+1} \int_{[0,\varepsilon]} (b_p(\varepsilon) - b_p(r_1))^k b'_p(r_1) d\mu(r_1) \tag{55}$$

$$= n \int_M d\nu(p) \sum_{k=0}^{n-1} \binom{n-1}{k} (1 - b_p(\varepsilon))^{n-1-k} \frac{1}{k+1} \frac{b_p(\varepsilon)^{k+1}}{k+1} \tag{56}$$

$$= n \int_M d\nu(p) \sum_{k=0}^{n-1} \binom{n-1}{k} \frac{1}{(k+1)^2} (1 - b_p(\varepsilon))^{n-1-k} b_p(\varepsilon)^{k+1} \tag{57}$$

$$= \int_M d\nu(p) \sum_{j=1}^n \binom{n}{j} \frac{1}{j} (1 - b_p(\varepsilon))^{n-j} b_p(\varepsilon)^j, \tag{58}$$

where the last is performed by re-indexing $j = k + 1$ and applying the property of the binomial coefficient $\binom{n-1}{j-1} = \frac{j}{n} \binom{n}{j}$. Applying Lemma 2, we rewrite the result in a more convenient form

$$\int_M \sum_{k=1}^n \frac{(1 - b_p(\varepsilon))^{n-k} - (1 - b_p(\varepsilon))^n}{k} d\nu(p). \tag{59}$$

and summing Equation (52) with Equation (59), we get our final expression

$$p_S^n(\varepsilon) = \mathbb{E}_{p \sim \nu} \left[ \frac{1}{n} + \sum_{k=1}^{n-1} \frac{(1 - b_p(\varepsilon))^{n-k} - (1 - b_p(\varepsilon))^n}{k} \right].$$

### A.5.2 IDENTIFICATION TASK

Re-tracing the first steps of Appendix A.5.1 and Appendix A.3.2

$$\mathbb{P}(X = Y) = n \int_M \sum_{p \in \{x_1, \dots, x_n\}} \frac{1}{n} \mathbb{P}(Y = 1 | X = 1, (x_1, \dots, x_n, p)) \mathbb{1}[X(x_1, \dots, x_n, p) = 1] d\nu(x_1) \cdots d\nu(x_n) \tag{60}$$

$$= \int_M d\nu(x_1) \int_{M^{n-1}} \frac{g(x_1, x_1)}{g(x_1, x_1) + \sum_{i=2}^n g(x_i, x_1)} d\nu(x_2) \cdots d\nu(x_n) \tag{61}$$

$$= \int_M d\nu(x_1) \int_{(0, \infty]} dS_p(r_2) \cdots \int_{(0, \infty]} dS_p(r_n) \frac{1}{1 + \sum_{i=2}^n g(r_i)}. \tag{62}$$

Just like we saw in the proof of the similarity task, here any stimulus different from the probe will contribute to the denominator of the decision function if and only if it falls in $B_\varepsilon(x_1)$. Moreover, the decision function depends only on the number of such stimuli and not on which ones contribute to the denominator. Therefore, we can write

$$\mathbb{P}(X = Y) = \int_M d\nu(x_1) \sum_{k=0}^{n-1} \binom{n-1}{k} b_{x_1}(\varepsilon)^k (1 - b_{x_1}(\varepsilon))^{n-1-k} \frac{1}{k+1} \tag{63}$$

$$= \int_M d\nu(x_1) \frac{j}{n} \sum_{j=1}^n \frac{1}{j} \binom{n}{j} b_{x_1}(\varepsilon)^{j-1} (1 - b_{x_1}(\varepsilon))^{n-j} \tag{64}$$

$$= \mathbb{E}_{p \sim \nu} \left[ \frac{1 - (1 - b_p(\varepsilon))^n}{n b_p(\varepsilon)} \right], \tag{65}$$

where we used the property of the binomial coefficient $\binom{n-1}{j-1} = \frac{j}{n} \binom{n}{j}$.

### A.6 PROOF OF PROPOSITION 1

We want to compute $p_S$ and $p_I$ for the uniform measure on the flat circle $M = [0, 1]$ with $d(x, y) = \min(|x - y|, 1 - |x - y|)$ for the linearly decaying similarity function with resolution $\varepsilon$, $g(r) = \sigma\left(1 - \frac{r}{\varepsilon}\right)$, where $\sigma(x) = \max(x, 0)$.

First, note that for the uniform measure, we have that

$$b_x(\varepsilon) = \nu(B_\varepsilon(x)) = \begin{cases} 2\varepsilon & \text{if } \varepsilon \leq \frac{1}{2} \\ 1 & \text{if } \varepsilon > \frac{1}{2} \end{cases} = b(\varepsilon),$$

i.e. the length of the interval $[-\varepsilon, \varepsilon]$ on the circle. Accordingly, we have that the measure $S_x$ is such that

$$S_x(E) = S(E) \int_E b'(r) d\mu(r) = 2\mu(E),$$

if $E \subseteq [0, \frac{1}{2}]$.

**Similarity task.** We start from Equation (9) and, once again, consider the different cases. If $r_1, r_2 > \varepsilon$, there is no difference from the constant case: the probe has similarity 0 with both $x_1$ and $x_2$, therefore the model is maximally uncertain. This term will contribute $(1 - b(\varepsilon))^2/2$ to $\mathbb{P}(Y = X)$.

If $r_1 \leq \varepsilon$ and $r_2 > \varepsilon$, there is no difference from the constant case as the probe is similar to $x_1$ with no interference from $x_2$. We get a contribution of $2b(\varepsilon)(1 - b(\varepsilon))$.

If $r_1 \leq \varepsilon, r_2 \leq \varepsilon$, we need to compute

$$2 \int_{[0, \varepsilon]} dS(r_1) \int_{(r_1, \varepsilon]} dS(r_2) \frac{g(r_1)}{g(r_1) + g(r_2)} = 8 \int_{[0, \varepsilon]} \int_{(r_1, \varepsilon]} \frac{1 - r_1/\varepsilon}{1 - r_1/\varepsilon + 1 - r_2/\varepsilon} d\mu(r_1) d\mu(r_2) \tag{66}$$

$$= 8 \int_{[0, \varepsilon]} (\varepsilon - r_1) \log(2) d\mu(r_2) = 8 \cdot \frac{1}{2} \varepsilon^2 \log(2) = (2\varepsilon)^2 \log(2) = \log(2) b(\varepsilon)^2. \tag{67}$$

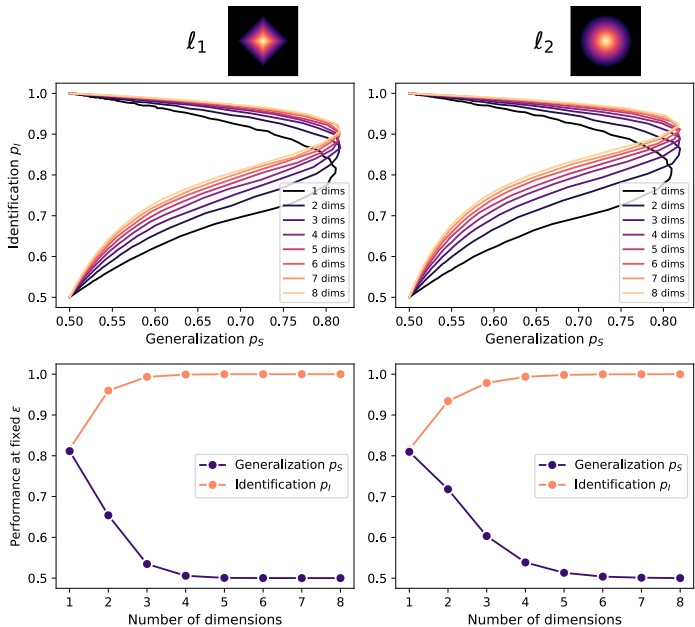

Figure 6: **Top row.** Generalization $p_S$ and identification $p_I$ accuracy curves for inputs sampled from a higher-dimensional torus equipped with $\ell_1$ (left column) and $\ell_2$ distance metric (right column) and linearly decaying similarity functions (depicted above). **Bottom row.** Generalization and identification accuracies as functions of the number of dimensions, for $\epsilon$ fixed to the maximum-achieving resolution of $p_S$ on the 1-dimension torus.

Putting together the three contributions, we get

$$\mathbb{P}(Y = X) = \frac{1}{2} - b(\varepsilon) + \frac{1}{2}b(\varepsilon)^2 + 2b(\varepsilon) - 2b(\varepsilon)^2 + \log(2)b(\varepsilon)^2 = \frac{1}{2} + b(\varepsilon) - (3/2 - \log(2))b(\varepsilon)^2.$$

**Identification task.** We start from Equation (22) and consider two cases. If $r > \varepsilon$, then $x_2$ does not interfere with the probe and thus the model will choose $x_1$ with probability $1$. Just like the constant case, we get a contribution of $1 - b(\varepsilon)$.

If $r \leq \varepsilon$, we need to compute

$$\int_{(0,\varepsilon]} \frac{1}{1 + g(r)} dS(r) = \int_{(0,\varepsilon]} 2\frac{1}{1 + 1 - r/\varepsilon} d\mu(r) = 2\log(2)\varepsilon = \log(2)b(\varepsilon). \tag{68}$$

In total, we get

$$\mathbb{P}(Y = X) = 1 - b(\varepsilon) + \log(2)b(\varepsilon) = 1 - (1 - \log(2))b(\varepsilon).$$

## A.7 DIMENSIONALITY ANALYSIS

Here we numerically check how the results change when the input space is multi-dimensional. We consider a $d$-dimensional (flat) torus whose points are $d$-tuples $x = (x_1, \ldots, x_d) \in [0,1]^d$. Each coordinate thus lives on a circular space. Following classical work describing similarity in multi-dimensional spaces Nosofsky (1986), we consider the Minkowski metric on such space

$$d(x, y) = \left(\sum_{i=1}^{d} \min(|y_i - x_i|, 1 - |y_i - x_i|)^p\right)^{1/p} \tag{69}$$

with $p = 1, 2$ and an agent implementing linearly decaying similarity functions $g(r) = \max(1 - r, 0)$ with respect to that metric.

In the top row of Figure 6, we show $p_S, p_I$ curves as the dimension $d$ of the space increases from 1 to 8. We see how, for both $\ell_1$ and $\ell_2$ metrics, the curve shift towards higher identification performances

while staying approximately constant in generalization. Fixing a single resolution value $\varepsilon$ and checking how performances depend on dimension (bottom row of Figure 6) shows fast decaying generalization and fast increasing identification. We can interpret this as an effect of the curse/blessing of dimensionality. In higher-dimensional spaces the volume covered by the similarity function (whose "radius" $\varepsilon$ is fixed) decreases. This results in a reduction in interference between representations, i.e. higher identification capabilities but, at the same time, the similarities are not wide enough to support generalization.

## A.8 DETAILS ON NUMERICAL EXPERIMENTS

All the code used to produce the results can be found in `https://anonymous.4open.science/r/generalization-7155`.

### A.8.1 TOY MODEL

The architecture of the toy model we used is the following linear bias-less autoencoder with a nonlinearity at the end

$$f(x) = \sigma(W^\top W x),$$

where $\sigma$ is the ReLU activation function $\sigma(x) = \max(x, 0)$ and $W \in \mathbb{R}^{m \times l}$.

In both the pure-reconstruction and semantic experiments, the inputs were chosen to be $l = 50$ one-hot vectors $x = e_i \ \forall i = 1, \dots, m$. The hidden space dimension was chosen to be $m = 10$.

The pure reconstruction experiment is performed by minimizing the MSE loss between input one-hot and its reconstruction through the network

$$L_{\text{rec}} = \sum_{i=1}^{l} \left\| e_i - \sigma(W^\top W e_i) \right\|^2 = \sum_{i=1}^{l} \left\| e_i - \sigma(W^\top w_i) \right\|^2.$$

In the semantic case, the loss is built in the following way. Three different indices $i, j, k$ are picked randomly and their associated one-hots are built $e_i, e_j, e_k$. Then, we compute the ratio of similarities

$$D_i = \frac{\sigma(w_i^\top w_k)}{\sigma(w_i^\top w_k) + \sigma(w_j^\top w_k)}, \ D_j = \frac{\sigma(w_j^\top w_k)}{\sigma(w_i^\top w_k) + \sigma(w_j^\top w_k)}.$$

The index $\hat{i} \in \{i, j\}$ of the correct answer is computed by taking the minimum between $d(x_i, x_k)$ and $d(x_j, x_k)$, where the distance function is given as a training input in the form of a distance matrix. The loss, finally, is computed by taking the Negative Log Likelihood Loss (NLL) between the distribution $(D_i, D_j)$ and the one hot vector encoding the correct response.

$$L_{\text{sim}} = -\frac{1}{2} D_{\hat{i}}.$$

For all experiments, each epoch is made of 2000 samples, with batch size 128. The models are trained for 500 epochs with the Adam optimizer, with learning rate 0.0007 and 0 weight decay.

Given that random vectors in high-dimensional space tend to be close to orthogonal, biasing the model towards high $p_I$, we initialize the weight matrix with i.i.d. uniform in the interval $[0, 2]$.

At each epoch, the model is evaluated by performing similarity and identification tasks. 1,000 triplets $(i, j, k)$ ($k \in \{i, j\}$ for the identification) are extracted, and the average $D_{\hat{i}}$ is recorded to obtain the values of $p_S$ and $p_I$ shown in Figure 4. The average similarity functions shown in the figure's insets are obtained as $g_i(j) = \sigma(w_i^\top w_j)$ for every $j \in 1, \dots, l$. Leveraging the symmetry of the circular structure, each vector $g_i$ is circularly shifted so that the index $i$ goes to the center of the circle $g_i \mapsto \tilde{g}_i$. Finally, we take the average over $i$, $\tilde{g} = \frac{1}{l} \sum_{i=1}^{l} \tilde{g}_i$.

We show the distance matrices for the circle and line experiments, together with the full learned similarity matrices for a single run in Figure 7.

In Figure 8, moreover, we see the results of the three different trainings for three values of the neural network's latent dimension. As it increases, we see how the model is able to have less interference

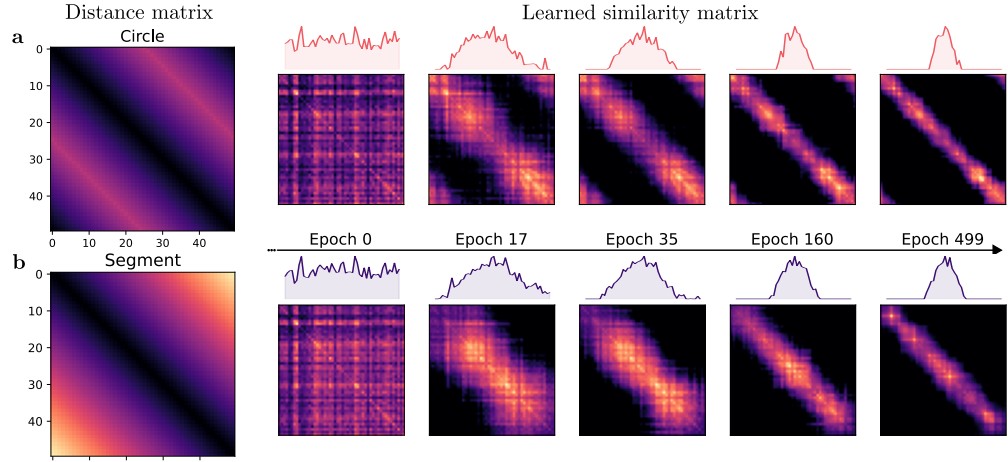

Figure 7: Visualization of the distance matrix (left) and the learned similarity matrices through training for the circle (top row) and the segment (bottom row).

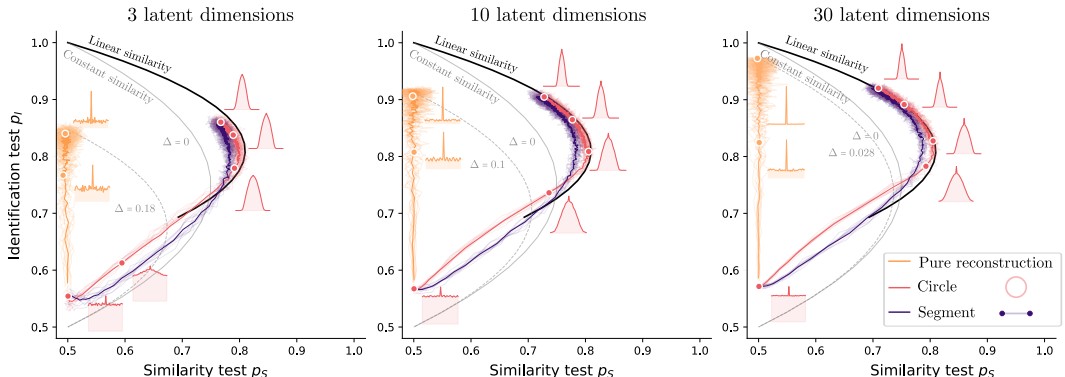

Figure 8: Different training trajectories of the toy model with different latent dimensions, visualized as in Figure 4.

between representations, signified by $p_I$ being able to reach higher values. Visualizing the average learned similarity functions and estimating the noise value, we are able in all cases to predict the maximum $p_I$ using Theorem 2.

### A.8.2 CONVOLUTIONAL NEURAL NETWORK FINE-TUNED ON EVOLUTIONARY DISTANCES AMONG BIRD SPECIES

**Experimental setup.** To test our theoretical predictions in a realistic computer vision setting, we fine-tuned a ResNet-50 model (He et al., 2016) pre-trained on ImageNet. We used the Caltech-UCSD Birds-200-2011 dataset (Wah et al., 2011), which contains 11,788 images of 200 bird species, paired with evolutionary distance data from the TimeTree database (Kumar et al., 2022). The experimental design involved two tasks with a consistent triplet-based evaluation format:

- **Identification task:** Given images of two reference species $(x_1, x_2)$ and a probe image, determine which reference species the probe belongs to.
- **Similarity task:** Given images of two reference species $(x_1, x_2)$ and a probe species $(p)$, determine which reference species is evolutionarily closer to the probe.

Using a contrastive loss that encouraged embedding bird images closer to their evolutionary relatives, we fine-tuned the model using a composite loss:

$$\mathcal{L} = (1 - \alpha)\,\mathcal{L}_{\text{id}} + \alpha\,\mathcal{L}_{\text{sim}},$$

where $\mathcal{L}_{\mathrm{id}}$ is a cross-entropy loss for species identification, and $\mathcal{L}_{\mathrm{sim}}$ aligns the embedding space with evolutionary distances. The parameter $\alpha$ controls the balance between identification and generalization objectives. During evaluation, we defined similarity using a threshold $\epsilon$ on feature distances, where distances below $\epsilon$ indicated similarity. This allowed us to systematically study the generalization-identification tradeoff by varying both $\alpha$ and $\epsilon$.

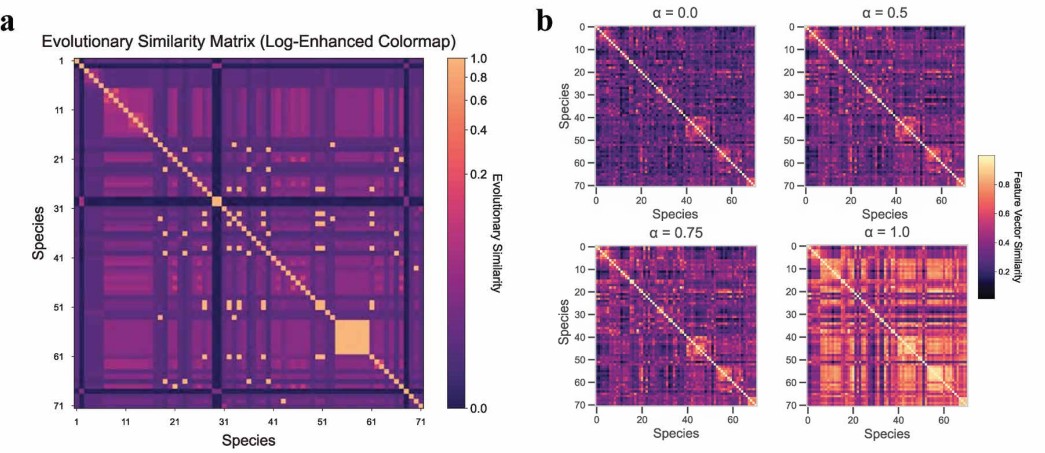

Figure 9: Evolutionary similarity between species obtained from (a) bird phylogeny and (b) the feature vector similarities as $\alpha$ is tuned.

**Training details.**   We trained the model for 15 epochs using SGD with momentum 0.9, weight decay $1e-4$, and an initial learning rate of 0.001, reduced by a factor of 0.1 when validation performance plateaued. To handle GPU memory constraints, we used a batch size of 8 with gradient accumulation over 4 steps (effective batch size 32). We tested $\alpha$ values ranging from 0.0 to 1.0 with several random seeds (42-46) to ensure robust results.

The birds dataset was split 64-16-20% for training, validation, and testing, with an additional 15% of species held out completely as out-of-distribution test data. The evolutionary distance loss ($\mathcal{L}_{\mathrm{sim}}$) was implemented by computing pairwise distances in feature space and aligning them with normalized evolutionary distances derived from the phylogenetic tree. This explicitly encouraged the CNN to map visual features into a space that preserved evolutionary relationships as shown in Figure 9.

**Theoretical connections.**   Our experimental framework directly maps to the theoretical constructs in Miller's Law. The identification task measures $p_I$ (probability of correct identification), while the similarity task measures $p_S$ (probability of correct similarity judgment). The threshold $\varepsilon$ corresponds to the resolution parameter in our theoretical framework, controlling the ball measure $b(\varepsilon)$ that determines which items are considered similar.

**Evolution during training.**   We monitored how the identification-generalization constraints evolved during training by tracking both scores across epochs. With $\alpha = 0$ (pure identification objective), models rapidly optimized for identification at the expense of generalization. As $\alpha$ increased, especially beyond 0.5, models traced distinct trajectories through $(p_s, p_I)$ (or G-I) space, with higher $\alpha$ values showing earlier and more pronounced shifts toward generalization.

The final equilibrium position in G-I space was primarily determined by $\alpha$, with higher $\alpha$ values reliably producing models with better generalization capabilities. Out-of-distribution testing revealed that models with higher $\alpha$ values demonstrated substantially better generalization to unseen bird species, confirming that the similarity-based training objective promotes more robust feature learning that captures fundamental biological relationships rather than superficial correlations.

**Results.**   As shown in Figure 5a, the bird CNN exhibits a clear tradeoff between generalization and identification. We expand these results in Figure 10, in which (a) shows how the G-I tradeoff is

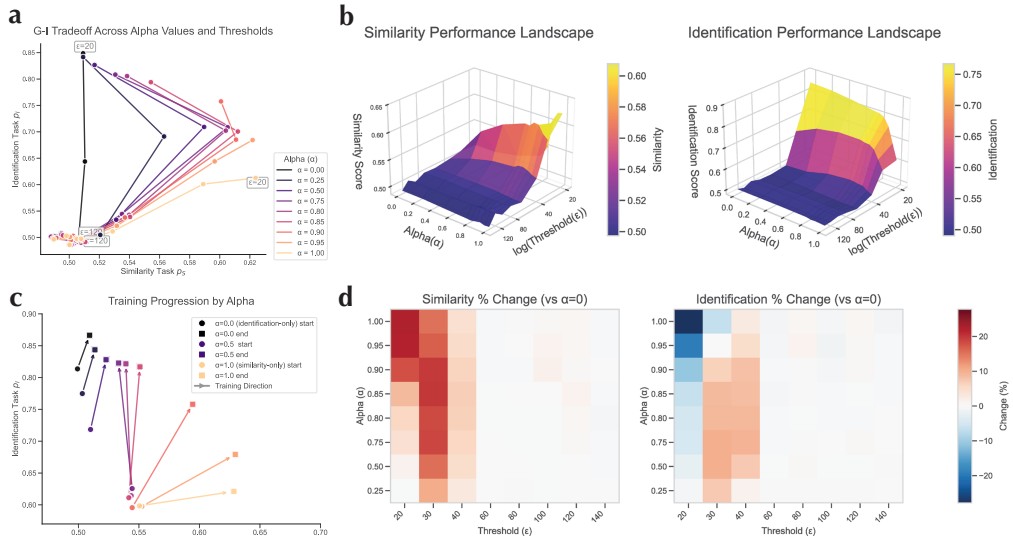

Figure 10: (a) Generalization-identification tradeoff parametrized by threshold resolution ($\varepsilon$) for various $\alpha$ values, showing how different regularization strengths shape the Pareto frontier. (b) Performance landscapes showing generalization (*left*) and identification (*right*) as continuous functions of $\alpha$ and $\log(\varepsilon)$. (c) Training trajectories in the G-I performance space for different $\alpha$ values, showing how higher $\alpha$ produces increasingly horizontal progressions that prioritize generalization over identification during learning. (d) Percentage change in generalization (*left*) and identification (*right*) relative to the pure reconstruction baseline ($\alpha = 0$), revealing a critical range of $\varepsilon$ values where positive deviations occur for both tasks.

parametrized by threshold resolution $\varepsilon$ for different $\alpha$ values, with each curve tracing the performance as $\varepsilon$ varies while $\alpha$ remains fixed. Panel (b) presents the full performance landscapes, showing generalization (*left*) and identification (*right*) scores as continuous functions of both $\alpha$ and $\log(\varepsilon)$. Panel (c) illustrates the training dynamics: as $\alpha$ increases, the training trajectories become increasingly horizontal, indicating that the learning process prioritizes generalization improvements over identification accuracy. Panel (d) quantifies the performance deviations from the pure reconstruction baseline ($\alpha = 0$) through heatmaps. For low thresholds ($\varepsilon \leq 20$) and high thresholds ($\varepsilon \geq 80$), deviations remain minimal across all $\alpha$ values. However, for intermediate thresholds ($30 \leq \varepsilon \leq 50$), we observe a critical regime: higher $\alpha$ values yield substantial positive deviations in generalization (up to 25% improvement) while identification shows moderate negative deviations (typically 10-20% decrease), most pronounced in the $\alpha \geq 0.8$ range. Notably, there exists a narrow band around $\varepsilon = 40$ where both landscapes show positive deviations for moderate $\alpha$ values, confirming our theoretical prediction that optimal threshold selection enables simultaneous enhancement of both generalization and identification beyond the pure reconstruction baseline.

### A.8.3 LLMs PERFORMING DATE-OF-BIRTH IDENTIFICATION VS SIMILARITY TASK

**Evidence of resolution**

**Experimental setup.** We investigated whether large language models exhibit semantic resolution when processing time information. We tested three models: gemma-2-2b-it (Team et al., 2024), Llama-3.2-3B-Instruct (Grattafiori et al., 2024), Qwen2.5-7B-Instruct (Bai et al., 2023) on the following task. The models are fed the system prompt `"You are a useful chatbot assistant."` and are asked to respond to the prompt `"A was born in x. B was born in y. Who was born closest to p? Answer with a single name."`

The variables `A,B,x,y` and `p` are generated in the following way:

1. A central year `c` is sampled uniformly from the set of integers $\{1500, 1501, \ldots, 1699\}$;

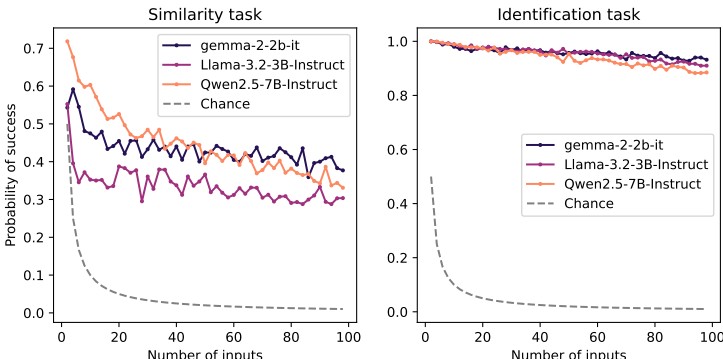

Figure 11: Similarity and identification performances of three LLMs on the interval of years $[800, 1599]$

2. For each value $\delta x \in \{20, 50, 100, 200\}$, we fix $\mathtt{x} = \mathtt{c} - \delta x$ and $\mathtt{y} = \mathtt{c} + \delta x$ with probability 0.5 and $\mathtt{x} = \mathtt{c} + \delta x$ and $\mathtt{y} = \mathtt{c} - \delta x$ with probability 0.5.

3. For each pair $\mathtt{x}, \mathtt{y}$ chosen in this way, we run the prompt with every $\mathtt{p} = \mathtt{c} + \delta\mathtt{p}, \delta\mathtt{p} \in \{\mathtt{c} - 300, \mathtt{c} + 300\}$.

4. The prompt with each value of $\mathtt{p}$ is ran 20 times, randomizing the variables $\mathtt{A}$ and $\mathtt{B}$, which are two different names sampled from the list `[ "Alice", "Bob", "Charlie", "David", "Eve", "Frank", "Grace", "Heidi", "Ivan", "Judy", "Karl", "Liam", "Mallory", "Nina", "Oscar", "Peggy", "Quentin", "Rupert", "Sybil", "Trent", "Uma", "Victor", "Walter", "Xander", "Yvonne", "Zach", "Abigail", "Benjamin", "Catherine", "Daniel", "Elena", "Frederick", "Gabriella", "Henry", "Isabella", "Jack", "Katherine", "Lucas", "Mia", "Nathan", "Olivia" ]`.

If the answer of the model belongs to the set of sampled names, then we check whether it is equal to the name associated to the smallest year.

We repeat the process, sampling $\mathtt{c}$ 40 times and averaging the results. What we obtain is a function from the probe displacement w.r.t. $\mathtt{c}$, $\delta\mathtt{p} \in [-300, 300]$ to the probability of the model decision function $\mathbb{E}_c[D_1(\mathtt{x}, \mathtt{y})] \in [0, 1]$.

**Results.** Figure 5b in the main text displays the probability of correct answers for both models across date displacements. Several observations support our theory:

1. Both models show high performance when probe dates are near reference dates (small displacements), but performance degrades as displacement increases.

2. The pattern follows our theoretical assumptions: the performance approaches chance level (0.5) when the reference years are close and the probe falls between them and when the reference years are far and probes is far form both.

**Similarity and identification tasks**

**Experimental setup.** We then performed similarity and identification tasks to gauge the performance of these three models as the number of inputs provided increases.

For each number of inputs $n \in \{2, 4, \ldots, 100\}$, we sample 1,000 prompts built in the following way

- **Similarity task:** `"A1 was born in x1. A2 was born in x2."` $+\cdots+$ `" An was born in xn."` `Who was born closest to p? Answer with a single name.`

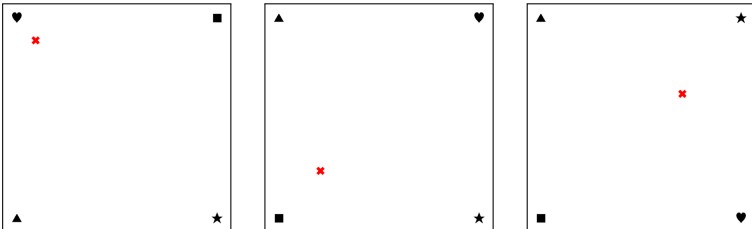

Figure 12: Examples of image inputs for the spatial resolution task.

- **Identification task:** `"A1 was born in x1.  A2 was born in x2."`+⋯+`"` `An was born in xn." Who was born in p?  Answer with a single name.`

Here `A1,...,An` are names sampled from a list of 200 names similar to the one described above, and `x1,...,xn` are random integers in the interval $[800, 1599]$. The probe `p` is a random integer in the same interval for the similarity task and, for the identification tasks, it is randomly chosen from the set $\{x1,...,xn\}$.

**Results.**    In Figure 11, we plot the performances obtained for the three models. Overall, we see that all models perform well on the identification task, with its performance decreasing with a small rate. Instead, for the similarity task we see how the performances are definitely worse, never being greater than 0.7 but decreasing in a much graceful way than the $1/n$ of random chance.

Interestingly, if we focus on Gemma and Qwen, we are able to qualitatively observe the same behavior of the theoretical model in Figure 3 of the main text. In fact, it appears that Qwen is favouring generalization, resulting in a good similarity task performance for a low number of items but a steeper decrease for increasing $n$. Gemma, instead, achieves close to chance similarity task performances when $n$ is small but decreases less rapidly when $n$ increases. If we map this to our theoretical investigation, it appears that Gemma is adopting a smaller $\varepsilon$ than Qwen, a conjecture which is corroborated by the identification test performance decreasing faster for the latter.

### A.8.4    VLM TASKS

To assess the presence of finite semantic resolution in vision–language models (VLMs), we designed several spatial similarity/identification tasks using synthetic images. Two VLMs were evaluated: `gemma-3-12b-it` and `Qwen2.5-VL-7B-Instruct`. Besides collecting the models' textual responses, we also logged the scores (logits) of selected, task-depending tokens, to inspect how each model ranks different token choices before softmax.

**Evidence of resolution**

**Dataset.**    We generated 1,000 images, each featuring four black stencils positioned at fixed locations on a white background. The stencils, chosen randomly between square, triangle, heart, star, varied specific position across images. Additionally, each image included a randomly placed red X, designated as the "target" (see Figure 12). We logged the distance between the target and each stencil.

**Experimental setup.**    We showed each image to both Qwen and Gemma, together with a query prompt: `"The picture contains four black shapes:  a square, triangle, heart and a star.  There is also a red X. Which black shape is the closest to the X? Respond with only the shape's name."`. We logged the models' textual responses and the token scores for *square*, *triangle*, *heart* and a *star*. For each sampled location, we recorded the model's output and computed an accuracy map. A smoothed version of this map was obtained by averaging over local neighbourhoods to reveal confidence gradients.

**Results.**    Figure 5c (main text) shows the accuracy maps for both models. In both cases, we observe a central region around each shape where the model is consistently correct, surrounded by

transition zones where accuracy rapidly deteriorates. This behavior mirrors the emergence of a finite resolution scale: when the red cross is placed sufficiently far from all reference shapes, the models are increasingly unable to resolve which object is closest.

Moreover, the spatial structure of the confusion regions reveals differences between models: `gemma-3-12b-it` exhibits a tighter high-confidence core, while `Qwen2.5-VL-7B-Instruct` shows broader transition bands, suggesting differences in their spatial encoding fidelity. The smoothed accuracy maps further support the hypothesis that VLMs implement a distance-dependent proximity function with finite support, analogous to the semantic similarity functions described in the theoretical model (Section 2).

**Color similarity task**

**Dataset.**    We created 5,000 images, each containing between 4 and 12 colored squares. Each square was labeled with a unique letter, serving as an identifier, as shown in Figure 13. The colors of the squares were generated using the HSV color model, where the hue (H) was assigned randomly, while saturation (S) and value (V) were maximized to ensure vivid and bright colors.

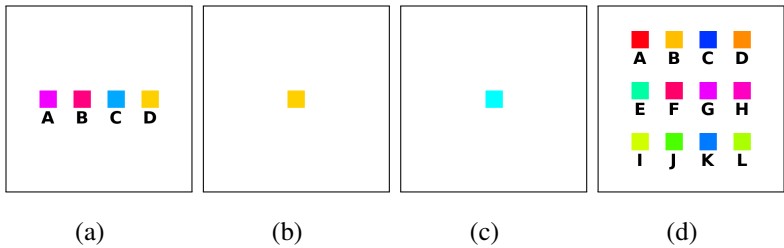

|  (a)  |  (b)  |  (c)  |  (d)  |

Figure 13: Examples of image inputs for the color similarity task. Panels (a) and (d) represent two reference images, with four and twelve colours respectively. Given the reference image (a), a query image for the identification task (color occurring in the reference image) is depicted in panel (b), and a query image for the similarity task (color not occurring in the reference image) is depicted in panel (c).

**Experimental setup.**    In this task, we presented the models with a pair of images and a textual query. The first image, dubbed *reference* image, contained 4 to 12 labelled color squares, as described above. The second image (*query image*) displayed a single, centered square, whose color was either one of the colors occurring in the reference image (identification task) or a completely random one (similarity task). In both case, the query was: `"In the first picture there are squares of different colors, labelled with uppercase letters.  the second picture there is only one target square.  Identify which square in the first picture is most similar to the target square in the second picture.  Reply with the corresponding letter and nothing else."`. We logged the color of the target square and its similarity with respect to all colors occurring in the paired reference image. We also logged the model's textual answers and the token score for each single letter (possible answers).

**Results.**    In Figure 14a we show the similarity task and identification task performances as functions of the number of input colored squares. In particular, we observe a decreasing identification performance which, in both models, can be fitted using the theoretical curve of Theorem 3 (main text). The fitted parameter $b(\varepsilon)$ suggests the presence of a larger effective resolution for Gemma and a lower one for Qwen.

To investigate this resolution, we gather, for each experiment, the scores each model assings to the letters associated to the *wrong* colors (thus excluding the most similar ones), together with their circular hue distance from the probe color, normalized to $[0, 0.5]$. We only take the scores associated to the wrong colors in order to avoid the bias of the correct answer having always low distance.

We plot the model score as a function of the distance in Figure 14b. Both model display an emergent resolution, with points with large hue distance being concentrated around a fixed "noise" level.

Figure 14: **a.** Similarity and identification probabilities for the color test explained in Appendix A.8.4. In the identification plot, the dashed curves are the theoretical curves of Theorem 3 (main text) fitted to the data. **b.** Token score associated to the *wrong* responses, as a function of the hue distance from the probe color. The dotted black lines represent the resolution values $b(\varepsilon)/2$ fitted using the theoretical resul of Theorem 3 (main text) on the identification performances.

Moreover, the scores for Gemma display a step-like shape suggesting that the learned similarity function may be similar to the constant similarity assumed in the theoretical analysis. Qwen, instead, shows a more continuous decrease in score-similarity with distance, more in line with the results obtained in Appendix A.8.3, and associated to higher performances (Figure 14a).

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
