# OpenReview forum: "Bound by semanticity: universal laws governing the generalization-identification tradeoff"
_ICLR.cc/2026/Conference — ICLR 2026 Poster_

### Official Review · Reviewer_NGhr · 2025-10-19

**Soundness:** 4
**Presentation:** 4
**Contribution:** 4
**Rating:** 10
**Confidence:** 3

**Summary:**

The authors study the tradeoff between generalization and identification through a notion of finite semantic resolution. They support their theory through a variety of convincing empirics in both a toy neural network and real-world models.

**Strengths:**

The paper is very well written overall. The exposition is clear, precise, and convincing. The theory is impressively predictive of both synthetic and real-world empirics. The central role of semantic resolution and its influence on the generalization/identification makes for a very compelling story. Well done overall!

**Weaknesses:**

See Questions below.

**Questions:**

For your empirics with the toy neural network model, do you have an idea why the model seems to learn a linearly decaying distance function? Since it sounds like the similarity task employed a conventional softmax output / cross-entropy loss, it seems like the natural distance function would be exponentially decaying?

You mention that increasing the decay rate in exponential distance will increase both generalization and identification. In your experiments with Transformers (which presumably employ an exponential distance hard-coded in softmax attention), did you find that increasing the decay rate boosts generalization and identification accordingly? Based on your prescriptions, to optimally employ these models, should we always increase decay rate at test time for maximal performance?

---

> ### Author Response · Authors · 2025-11-21
>
> ## Questions
>
> - *For your empirics with the toy neural network model, do you have an idea why the model seems to learn a linearly decaying distance function? Since it sounds like the similarity task employed a conventional softmax output/cross-entropy loss, it seems like the natural distance function would be exponentially decaying?*
>
> We thank the reviewer for the question and for their positive assessment of our work.
>
> In our toy neural network experiments, the learned similarity function decays approximately linearly with respect to the natural distance on the underlying input space (in this case, a circle).  Concretely, the network arranges the embeddings in latent space so that the cosine similarity between embeddings decreases roughly linearly with geodesic distance on the circle.
> Because of this geometric organization, the cross-entropy/softmax objective does not, by itself, impose an exponential decay profile on the learned similarity.
> There is no direct pressure favoring an exponential rather than a linear relationship between distance and similarity.
>
> A priori, one might indeed expect exponential similarity gradients, in part because they arise as optimal solutions under efficient coding assumptions such as rate–distortion theory (e.g., Sims, 2018).  However, in this particular task setting, we do not observe such exponential profiles in practice, and we do not yet have a clear theoretical explanation for why the trained toy models preferentially approximate a linear decay.
>
> We view this as an interesting open direction, but it does not affect the qualitative empirical finding that the learned similarity function displays finite resolution and supports the predicted tradeoff.
>
>
> - *You mention that increasing the decay rate in exponential distance will increase both generalization and identification. In your experiments with Transformers (which presumably employ an exponential distance hard-coded in softmax attention), did you find that increasing the decay rate boosts generalization and identification accordingly? Based on your prescriptions, to optimally employ these models, should we always increase decay rate at test time for maximal performance?*
>
>
> We thank the reviewer for this thoughtful question.
>
> First, we would like to clarify that in our theoretical analysis, the exponential similarity function refers to similarity as a function of the "true" natural distance in the space of inputs whose geometry the model is learning.
> The decay rate in this exponential similarity is therefore a property of the learned representation, not an explicit parameter we can directly manipulate. It is conceptually distinct from architectural parameters, such as the softmax temperature used inside transformer attention.
>
> Under this interpretation, the fact that increasing the decay rate of an exponential similarity (of the learned representations) boosts both generalization and identification, holds only in the idealized case where similarities can be computed with arbitrarily high precision.  As we discuss, once finite resolution or representational noise is introduced, this monotonic improvement breaks down: beyond a certain point, increasing the decay rate no longer helps and becomes detrimental.

---

> > ### Comment · Reviewer_NGhr · 2025-11-26
> >
> > Thanks for the additional clarification and elaboration. I continue to affirm my support for this manuscript. Well done!

---

### Official Review · Reviewer_1bfK · 2025-10-31

**Soundness:** 3
**Presentation:** 3
**Contribution:** 3
**Rating:** 8
**Confidence:** 3

**Summary:**

This paper proposes that neural systems with finite semantic resolution face a fundamental tradeoff between generalization (correctly judging similarity between different inputs) and identification (distinguishing exact inputs). The authors derive closed-form Pareto curves characterizing this tradeoff for systems that compare inputs via a decaying similarity function, showing that performance lies on a universal frontier independent of input space geometry. They extend the theory to cases with noise and multiple items, predicting a sharp $1/n$ capacity collapse in multi-input settings. They also provide extensive empirical evidence that semantic resolution acts as a general information constraint in complex systems

**Strengths:**

1. The central tradeoff (Thm 1) is formally derived under clear assumptions about similarity decay and finite resolution. The analytical Pareto frontier is a nice contribution: mathematically precise, easy to reason about, and interpretable in terms of task accuracy.

2. The empirical sections demonstrate that real-world models qualitatively follow the predicted tradeoff curves

3. this work links the G-I tradeoff to well-known cognitive constraints like binding failures, generalization gradients, and working memory, providing a cohesive narrative.

**Weaknesses:**

1. The core tradeoff is derived assuming specific forms of similarity decay and decision rules. Would the authors elaborate how universal or sensitive their conclusion is to the choice of decay function? In Discussion, the authors refer to “finite-resolution similarity” as a universal constraint, suggesting that the existence of the tradeoff is robust, even if the exact shape of the curve depends on the similarity decay. It is unclear whether in those larger models, the similarity function follows the linear decay or takes a different shape.

**Questions:**

Is there an optimal distance function? the paper motivates its choice of decaying similarity functions using Shepard’s law, stating that generalization should decay with distance. Prior work (e.g., Sims, 2018) has shown that such generalization gradients follow from efficient coding via rate-distortion theory. Would you clarify whether the similarity function is intended to model a learned or designed encoding independently of such optimization?

---

> ### Author Response · Authors · 2025-11-21
>
> ## Weaknesses
> We thank the reviewer for their positive assessment and for raising this important question.
>
> It is correct that our theoretical derivation establishes the tradeoff only for the simplified cases of *(i)* a constant similarity function with a hard cutoff and *(ii)* a linearly decaying similarity.
> These choices were motivated by analytical tractability: they allow us to obtain closed-form expressions and to clearly illustrate the underlying mechanism.
>
> Our motivation for analyzing the hard-cutoff case in particular was to demonstrate that the tradeoff persists even when all features of a similarity function, except its finite resolution, are removed.
> This isolates resolution as the essential ingredient, rather than any peculiarities of the similarity function.
> We agree with the reviewer that understanding the robustness of the result to general choices of similarity function is an important question, but we believe that a complete theoretical characterization of the effects of specific functional choices is beyond the scope of the present paper.
>
> Nonetheless, we believe the qualitative form of the tradeoff is robust. Two observations support this claim:
> - Neural network toy models learn similarity functions that are not strictly linear, but are well *approximated* by linear decay. Despite not being exactly linear nor constant, the models consistently exhibit the same tradeoff, suggesting that the precise functional form is not critical to the emergence of the tradeoff, and that the tradeoff itself is robust to different forms of similarity fucntions.
>
> - Independent evidence comes from Frankland et al. (2021), where the authors use exponential similarity functions (inspired by earlier results by Sims, 2018) with noise and still observe the same qualitative phenomenon.  This indicates that neither linearity nor the idealized hard cutoff is necessary for the tradeoff to emerge, but rather the existence of finite resolution.
>
> ## Questions
> It is true that exponential similarity functions are a common assumption in the literature, both because they fit behavioral data well and because they emerge as optimal solutions under efficient coding principles such as rate–distortion theory (e.g., Sims, 2018).
> However, our framework does not assume or depend on exponential decays specifically.
>
> In our work, the similarity function is not intended to represent an optimal or designed encoding, but rather the result of learning: the neural network, through training, arranges inputs in a representation space where similarities can be computed.
> Our assumption is that the effective precision of these similarities decreases with distance, such that items beyond the semantic resolution become effectively indistinguishable.
> To illustrate this in the simplest possible way, we model this resolution limit using constant similarity functions with a hard cutoff.
>
> Thus, the similarity function in our model should be viewed as an idealized abstraction of learned representational geometry.

---

### Official Review · Reviewer_8DWP · 2025-10-31

**Soundness:** 3
**Presentation:** 4
**Contribution:** 3
**Rating:** 6
**Confidence:** 3

**Summary:**

The paper investigates the tradeoff between representational fidelity and distinctness under "finite semantic resolution". It shows that if a model’s similarity function has finite “semantic resolution” ε, then its accuracy on generalization tasks (p_S) and on identification tasks (p_I) must lie on a universal Pareto front.

If a model's embedding/similarity has a finite-resolution "floor", we should expect identification to drop as one pushes for broader generalization, and vice versa. Handling n simultaneous items suffers a sharp ∼1/n collapse in identification capacity; i.e., multi-object reasoning should not scale linearly just by adding parameters.
One can choose ε (via architecture, temperature, or thresholding) to target the sweet spot where p_S is maximized (roughly when the “similarity ball” covers half the space).

In short, the paper provides a simple geometry+noise model one can use to set thresholds, pick temperatures, and anticipate how adding more items or pushing for “similarity-aware” training will impact identification.

**Strengths:**

Comprehensive background section with helpful literature review.

The authors take the time to carefully explain their setup, accompanying notation with helpful illustrative examples.

Authors tested both toy (allowing for theoretical analysis) and realistic models (allowing for confirmation at scale).

Text is well-written, figures are clear, elegant, and helpful.

Code is provided showing how to reproduce the results and figures used the paper.

**Weaknesses:**

1. As mentioned by the authors in Limitations subsection, compositional representations and hierarchy were outside the scope of the study. Adding a small pilot on a compositional task to show if/why the current theory breaks would strengthen the paper.


__Minor points__:

2. The concept of “semantic resolution” feels somewhat over-introduced. Mathematically, it appears equivalent to a kernel scale or bandwidth that simply controls how similarity decays with distance. While the term is evocative and may carry intuitive meaning across domains, its use risks adding unnecessary jargon. I suggest that the authors clarify whether “semantic resolution” represents a genuinely new construct (i.e., beyond a kernel bandwidth) or simply reinterprets that familiar notion in "semantic" terms. A short comparison or restatement using standard terminology would make the paper easier to follow for readers from machine learning backgrounds.

3. Although there is nothing technically wrong with the abstract, it is very dense and hard to unpack on a first read. The authors could make it a bit lighter in order to appeal to a broader, non-expert audience. For example, by simplifying a few long sentences and highlighting the main contribution more explicitly rather than embedding it deep within the paragraph.

Typos:
- Please fix the m-dashes in line 84.
- and [an] additional one (line 97).
- maximal unncertain[ty] (line 158).
- m-dash in line 189.
- m-dash in line 457.
- m-dashes in lines 464--465.
- please fix hyphen in line 481.

**Questions:**

How is "semantic resolution" estimated in practice, and does its value depend on how the embeddings are scaled or normalized? That’s important because, otherwise, "semantic resolution" might just be a unit-dependent artefact rather than a stable, interpretable quantity.

---

> ### Author Response · Authors · 2025-11-21
>
> ## Weaknesses
> 1. *On compositional representations*
>
> We thank the reviewer for the valuable suggestion.  We fully agree that extending the framework to compositional and hierarchical representations is an important next step.
> However, introducing compositionality into our setup is non-trivial: it requires specifying how individual feature dimensions combine, how similarity should operate over structured inputs, and how the model represents multi-part objects.
> We believe that a complete analysis would be beyond the scope of the present work.
>
> As a preliminary step in this direction, we conducted an exploratory analysis examining how the tradeoff behaves when inputs lie in a multi-dimensional latent space.  Concretely, we considered inputs sampled from an $n$-dimensional (flat) torus (thus composed of $n$-dimensional points) and asked how to define the "correct" distance function over dimensions.
> Following the classic treatment of multidimensional similarity in Nosofsky’s works (see e.g. [1]), we combined per-dimension distances using both $\ell_1$ and $\ell_2$ norms and studied the resulting similarity functions.
>
> Assuming linearly decaying similarity, we obtain a family of tradeoff curves that systematically shift as dimensionality increases.  In particular, fixing the resolution parameter $\varepsilon$, increasing the number of dimensions leads to higher identification and lower generalization.
> This pattern reflects a geometric effect: in higher dimensions the similarity function's resolution covers less volume, making items less likely to interfere and easier to distinguish (boosting identification), but generalization gradients decay too quickly relative to the typical inter-item distance (reducing generalization).
>
> We have added a short section and a corresponding plot (Fig. 6) in the supplementary material (Section A.7) summarizing this pilot analysis.  While this does not yet address full compositionality, it provides an initial indication of how the theory behaves when multiple dimensions are taken into account.
>
> [1] Nosofsky, Robert M. "Attention, similarity, and the identification–categorization relationship." Journal of experimental psychology: General 115.1 (1986): 39.
>
> 2. *On the concept of semantic resolution*
>
> We thank the reviewer for this helpful comment.
> We agree that our notion of semantic resolution is closely related to the idea of a kernel bandwidth, in that both control how similarity decays with distance.
> Our choice of terminology stems from the cutoff-based framework we analyze: in that setting, the bandwidth parameter directly determines the effective resolution of the similarity computation, establishing the scale beyond which things become indistinguishable.
> For this reason, we believe that the notion of “resolution” captures the operational role of the parameter more transparently in our context.
> We appreciate the reviewer’s suggestion and have added a brief clarification in the main text (L179) to explicitly relate semantic resolution to the standard kernel bandwidth terminology.
>
> 3. *On the abstract*
>
> Thank you for pointing out that, in its current form, the abstract may be hard to understand.
> We rewrote and simplified it in the updated document.
>
> ## Questions
> We thank the reviewer for raising this important point.
>
> In our work, semantic resolution is primarily introduced as a theoretical idealization: it represents the effective scale at which a model’s similarity computations can reliably distinguish between inputs.
> In practice, this quantity is implicit in the learned representations and depends on how the model organizes and processes embeddings when making decisions.
> It is therefore not something we fix or normalize directly.
>
> In the simplified “toy” settings where the structure of the input space is known (e.g., circular or linear stimuli), we can think of estimating semantic resolution in two complementary ways:
>
> *1)* As shown in Fig. 5b (and Fig. 14 in the appendix), we can infer semantic resolution solely from the model’s behavior on generalization or identification tasks.
> By fitting the shape of the empirical generalization curve or by measuring where performance transitions occur, we can estimate at what distance the model can no longer reliably discriminate inputs.
>
> *2)* Alternatively, one could directly probe the internal representations of the model.
> Under the linear representation hypothesis, where features correspond to directions in embedding space, computing the cosine similarity between embeddings as a function of the true distance between the corresponding inputs yields a generalization gradient akin to what wee assume.
> The point at which this similarity function flattens or collapses provides an estimate of the model’s semantic resolution.
>
> ## Typos
> We fixed the typos according to your suggestions. Thank you for pointing them out.

---

> > ### Comment · Reviewer_8DWP · 2025-11-27
> >
> > I thank the authors for their responses and clarifications. I believe their additions to the text make this manuscript even stronger so I am updating my recommendation to reflect this.

---

### Official Review · Reviewer_MQEJ · 2025-10-31

**Soundness:** 4
**Presentation:** 3
**Contribution:** 3
**Rating:** 6
**Confidence:** 4

**Summary:**

The authors derive a theory for how a representation (for constrained resolutions) trades off between generalization and identification, a long known phenomenon from the cognitive science literature. In particular, this theory implies a Pareto frontier between generalization and identification performance. They empirically demonstrate that ReLU networks navigate such a Pareto frontier and find similar performance in a CNN finetuned on a mixture of an identification and generalization task. Finally, they show that LLMs and VLMs both show evidence of a finite resolution, a key assumption of their theory.

**Strengths:**

This manuscript provides a well articulated contribution to the field, formalizing a tradeoff that had previously been empirically observed. The theory is well-presented and I think the ReLU experiments in particular provide helpful support for the existence of the noted tradeoff. I appreciated the detailed contextualization of this present work in the field. Finally, the paper is generally well-written and the figures are well-designed. Below are some additional parts of the paper I particularly liked:

- The equations (3) and (4) are simply and immediately make the tradeoff intuitive.
- I think it's very interesting that the ReLU networks show some emergent evidence of this tradeoff even though they are only trained on the similarity task, not the identification task.
- Figure 4 demonstrates a good match between their (modified) theory and empirical observations
- Their Proposition 1 demonstrates that this theory can extend beyond the (somewhat minimal) binary similarity measure case.
- The detailed explanation of the different regimes and Fig. 2 are very helpful.

**Weaknesses:**

As noted above, I liked this paper. My primary concern is that the experiments in section 5 provide rather limited evidence of this tradeoff. Your suggestion (and the suggestion of the prior literature) that this is a universal tradeoff would suggest that it should be apparent even in models that weren't trained explicitly on the identification and similarity task you're measuring. The fact that models become better at identification/similarity as you're varying the parameter prioritizing one or the other loss function is maybe not particularly surprising (though I agree that it demonstrates that there is a tradeoff between those two functions). Moreover, the LLM and VLM experiments don't demonstrate a tradeoff but rather just show that resolution is limited overall. I thought the ReLU experiment was more compelling in this direction, as it trained the ReLU networks only on the similarity task and still demonstrated this emergent tradeoff over different epochs. Unless I'm missing something, I think acknowledging this limitation would be important.

**Questions:**

- Would you expect an emergent tradeoff between identification and similarity performance if you only trained the CNN on either the identification or similarity task over different epochs (akin to the ReLU network case)?
- Similarly, would you expect such a tradeoff e.g. in different pretrained models?
- Could you discuss how your insights would apply to e.g. the tasks examined in Campbell et al. (2024)?

---

> ### Author Response · Authors · 2025-11-21
>
> ## Weaknesses
> We sincerely thank the reviewer for the thoughtful evaluation. We appreciate the recognition that the ReLU experiments provide compelling evidence of emergent tradeoffs.
> The concern that there is limited evidence of the tradeoff in our CNN, LLM, and VLM experiments is well-taken and we'd like to clarify a few points of concern.  In particular, to better address the reviewer's comment, we would like to disentangle some of the observations.
>
> First, as hinted above, we agree that not all the experiments we perform above provide the same evidence of the tradeoff. Indeed, the experiments were organized on purpose along an axis of increasing complexity of the model and decreasing control possible on our part.
>
> The first "toy model" is the one for which we have the most control (as mentioned by the reviewer) and we can explicitly show the emergence of the tradeoff when the model is trained on the semantic similarity loss $L_{\text{sim}}$ versus the pure reconstruction one $L_{\text{rec}}$ (Figure 4 in the main text, and 7-8 in the SI). In this simplified case, we could also explicitly recover the semantic similarity functions parametrized by the underlying distance between stimuli.
>
> The CNN experiments increase the complexity of the model by starting from a pre-trained model and fine-tuning it using a combined loss $L = (1-\alpha)L_{id}$ $+ \alpha L_{sim}$.  In this sense, our CNN experiments already contain the limits suggested by the reviewer above, as we trained the models for a set of values between $\alpha =0$ (pure identification) to $\alpha = 1.0$ (pure similarity).
> Figure 5a shows evidence that, as $\alpha$ increases (pushing the loss closer toward generalization), the model loses identification performances and gains in similarity.
> Both single-objective conditions ($\alpha=0,1$) exhibit similar tradeoff patterns between identification and generalization scores as training progresses.
> When $\alpha=0$, the networks improve identification while generalization score remains steady. When $\alpha=1$, the networks improve generalization while the identification score remains steady (Figure 10 in the appendix).
> Both epoch-wise training trajectories move from lower-left to upper-middle regions of the G-I plane.
> This is directly parallel to the ReLU case the reviewer found compelling as the tradeoff emerges from finite-resolution representations rather than the competing loss functions.  The mixed $\alpha$ experiments then show that we can control where the model settles on this inherent Pareto front.
> We can add focused plots isolating the results from $\alpha=0$ and $\alpha=1$ training trajectories to emphasize this behavior. We apologize this result was somewhat obscured in the current presentation.
>
> Finally, in the experiment with LLMs and VLMs we do not have the capacity to control directly neither the training nor the resolution. Their rational performances and gains in similarity of those experiments was to provide direct evidence of the presence of a finite resolution even in very complex models.
> That said, we did try to modify the natural internal resolution by running experiments with the same LLMs of section 5 by priming the model with variable ranges for $x_1$ and $x_2$ values before rerunning the experiments (e.g. "Expect dates in the range between $X_1$ and $X_2$. A was born in $x_1$. B was born in $x_2$. Who was born closest to $p$?" , with $X_{1,2}$ varying across multiple orders), but we did not find any observable change in the performances of the model.
> We expect there might be other ways to manipulate directly the resolution in complex models. We added a comment regarding this in the Limitations and as a possible direction for future work.

---

> > ### Author Response · Authors · 2025-11-21
> >
> > ## Questions
> >
> > - *Would you expect an emergent tradeoff between identification and similarity performance if you only trained the CNN on either the identification or similarity task over different epochs (akin to the ReLU network case)*
> >
> > We thank the reviewer for the opportunity to clarify this point. In the CNN experiments, the tradeoff still is an emergent phenomenon, even when we enforce a mixture of reconstruction and similarity loss terms via the parameter $\alpha$.  In Figure 10 (a and b panels) in the SI we now show explicitly that for any $0<\alpha<1$ we find a Pareto front parameterized by the resolution $\varepsilon$.
> > We can highlight two specific cases as mentioned in the question.
> > For pure reconstruction ($\alpha=0$), we find a trajectory in the G-I plane that is very close to that observed for the toy model in the case of pure reconstruction (orange lines in Figures 4 and 8).
> >
> > When we instead set $\alpha=1.0$, we train the model only on the similarity task, that is, we explicitly instruct the model to prioritize generalization and effectively ignore identification in the loss function. As shown in Figure 10a, we still find evidence of the tradeoff as a function of the resolution $\varepsilon$, even though the Pareto front curve stops early due to limits on the viable resolution parameters induced by the limited nature of the evolutionary tree and bird images dataset that we used.  As support of this, if we train the network **almost** exclusively on the similarity task ($\alpha=.90-.95$) we recover clear evidence of the Pareto front.
> >
> > Thus, the tradeoff does indeed arises spontaneously as soon as $\alpha>0$, that is, as soon as any generalization is required.  Hard-coding a measure of competition between the identification and  generalization terms via intermediate values of $\alpha$ does indeed modulate the shape of the Pareto front but it does not prevent it.
> > Rather, it allows us to systematically explore how different regularization strengths affect the generalization-identification balance by tuning various "knobs" ($\alpha$, $\varepsilon$) in a controlled experimental setting.
> >
> > From this angle, the toy ReLU network demonstrates the emergence of the tradeoff without the requirement of specifying a resolution: indeed, it shows exactly that a finite resolution appears also without regularization terms.  In other words, we do not control $\varepsilon$ in the ReLU model and the model itself evolves toward a value close to the one maximizing generalization.
> >
> > Summarizing, the CNN framework allows to directly explore $\varepsilon$ and thus the model serves as a mechanistic tool to understand the phenomenon's underlying principles and to identify critical parameters, while the ReLU model shows how the tradeoff can emerge from representational constraints alone.
> >
> > - *Similarly, would you expect such a tradeoff e.g. in different pretrained models?*
> >
> > Yes, we expect the essence of the tradeoff to persist in more complex pretrained models.
> > The tradeoff arises from a fundamental property of finite-resolution similarity, and our experiments with CNNs and vision–language models already indicate that it appears well beyond toy settings.
> >
> > However, pretrained models present an important practical limitation: their representational geometry (and therefore their resolution) is fixed by the pretraining process.
> > Unlike in our controlled toy models, we cannot vary the resolution parameter to trace out the full theoretical tradeoff curve.
> > As a result, while we can observe where a given model lies on the $(p_S,p_I)$ plane, we cannot systematically shift its resolution to reproduce the entire family of curves predicted by the theory.

---

> > > ### Author Response · Authors · 2025-11-21
> > >
> > > - *Could you discuss how your insights would apply to e.g. the tasks examined in Campbell et al. (2024)?*
> > >
> > > We thank the reviewer for the question. A formal application of our framework to the tasks studied in Campbell et al. (2024) would require extending the theory to compositional inputs, which is an important direction, but currently beyond the scope of this paper (see response to Reviewer 8DWP).  Nonetheless, several conceptual connections can already be drawn.
> > >
> > > First, although the tasks differ from ours, both counting and visual search fundamentally rely on the system’s ability to separate multiple items in a scene.
> > > Counting requires distinguishing each item from the others, and search requires isolating a target from distractors.
> > > In this sense, both tasks can be naturally mapped onto the identification side of the tradeoff we study.
> > >
> > > In our identification analysis, the key limiting factor is interference: when representations of different items are not orthogonal, similarity between them increases the chance of confusion.
> > > Compositional inputs introduce such similarity structure automatically, even when individual feature dimensions are unstructured.
> > > For example, in a color–shape space, items that share a feature (e.g., the same color) will be more similar than items that differ on both dimensions.
> > > This similarity structure facilitates generalization but necessarily introduces interference that impairs identification.
> > >
> > > Viewed through this lens, several effects observed by Campbell et al. align with our predictions.
> > >
> > > In the conjunctive search condition, the target shares features with the distractors, creating non-negligible similarity and therefore higher interference, leading to worse performance.
> > > In the disjunctive search condition, the target is far from the distractors in feature space, reducing similarity and thus interference, which makes the task easier.
> > > In the counting experiments, performance is higher on high-entropy screens, where objects vary more in their features. This corresponds to greater separation in feature space, reduced interference, and therefore more accurate identification of each item.
> > >
> > > Together, these observations are consistent with our core claim: interference arising from similarity structure limits identification, particularly in multi-item settings.

---

> > > > ### Comment · Reviewer_MQEJ · 2025-11-24
> > > >
> > > > Thank you for your response!
> > > >
> > > > I found the existence the emergent tradeoff with respect to different thresholds epsilon very helpful towards demonstrating the relevance of your theory for larger models. (For what it's worth I find that plot as compelling as Fig. 5a and think it would be a great addition to the main text.)
> > > >
> > > > Regarding other pretrained models: I'm curious if the broader frontier is something you could look at over the course of curricula or between pretraining and finetuning (e.g. using the Pythia models: https://github.com/EleutherAI/pythia).
> > > >
> > > > Overall, I think this is a cool paper and your response has left me more comfortable with your claims regarding larger models. I've updated my score to an 8.

---

### Author Response · Authors · 2025-12-01
**Summary of the rebuttal phase**

We thank the reviewers for their thoughtful and constructive engagement.
Overall, the submission received consistently positive assessments (initial scores 6, 6, 8, and 10), with reviewers highlighting the novelty of the theoretical contribution, the clarity of exposition, and the importance of formalizing a principled tradeoff between
identification and generalization under finite-resolution similarity.
Below, we summarize the main concerns and how we addressed them, for conciseness and easy of access.

### 1. Strength of empirical evidence (Reviewer MQEJ).
The reviewer questioned whether the tradeoff truly emerges in CNNs, LLMs, and VLMs,  beyond setups where the loss explicitly mixes identification and similarity.
In our response, we clarified that the CNN experiments *do* show an emergent tradeoff even without mixed losses.
We added explicit plots (now in the SI) for $\alpha = 0$ and $\alpha = 1$,  demonstrating epoch-wise trajectories analogous to the toy ReLU case.
For LLMs and VLMs, we explained that these experiments are intended to demonstrate  the presence of finite semantic resolution in large pretrained models, not to trace the full Pareto front, since their representational geometry cannot be directly controlled.
We acknowledged these limitations in the revised manuscript.


### 2. Extension to compositional settings (Reviewer 8DWP).
The reviewer requested a pilot analysis examining whether the theory extends to compositional or hierarchical inputs.
We explained that full compositionality requires substantial modeling choices and is beyond the present scope.
However, we added a preliminary multidimensional analysis (new Fig. 6 in the Appendix)  showing how the tradeoff behaves when inputs lie on an $n$-dimensional torus, combining per-dimension distances via standard $\ell_1$ and $\ell_2$ norms.
This reveals systematic shifts in the tradeoff with dimensionality and provides a foundation for future extensions.

### 3. Interpretation of semantic resolution (Reviewer 8DWP).
The reviewer noted that "semantic resolution" resembles kernel bandwidth.
We clarified in the main text that the concepts are closely related, but "resolution" more transparently captures the operational role of the parameter in our cutoff-based framework.
We further explained how semantic resolution can be estimated either from behavioral performance curves or from the geometry of learned embeddings.

### 4. Robustness to similarity functions (Reviewer 1bfK).
The reviewer asked how sensitive the tradeoff is to the choice of similarity decay.
We emphasized that the constant- and linear-decay models were chosen for tractability and to isolate finite resolution as the essential ingredient.
Empirically, toy neural networks learn similarity functions that are not perfectly linear yet still produce the same qualitative tradeoff.
We also noted that prior work (Frankland et al. 2021) qualitatively observes similar phenomena using exponential similarity gradients with noise, further supporting robustness.

### 5. Learned distance functions and exponential decay (Reviewer ngHR).
The reviewer asked why toy models learn approximately linear decay and whether the theoretical treatment of exponential decay implies that Transformers should benefit from increasing the softmax temperature or decay rate.
We clarified that in the toy networks, the linear gradient arises from the geometry of embeddings, and the softmax does not impose an exponential relationship with true input distance.
Moreover, the theoretical "decay rate" refers to similarity in the learned representation space and is conceptually distinct from softmax temperature.
Thus, our results do not prescribe architectural tuning in Transformers.

---

### Meta-Review · Area_Chair_73xb · 2026-01-04

**Summary:**

This work is motivated by psychology and studies the generalization-identification tradeoff in AI models. It shows that there is a tradeoff between generalization and identification when the model has a finite resolution. While the average score is high, I personally do not see the contribution as too significant. After all, I think the paper is saying a simple machine-learning message that everyone agrees with: at high "resolution" (which I feel is really just a parameter, almost identical to what people call "memorization"), the models distinguish all data and thus do not generalize. For this reason, I recommend a poster presentation

**Reviewer Concerns:**

Most of the concerns are minor and seem well addressed.

**Reviewer Scores:**

Given the minor problems, I do not see how they will change significantly

---

### Decision · Program_Chairs · 2026-01-26

Accept (Poster)